# Selection and the direction of phenotypic evolution

**François Mallard\*, Bruno Afonso, Henrique Teotónio\***

Institut de Biologie de l'École Normale Supérieure, CNRS UMR 8197, Inserm U1024, PSL Research University, Paris, France

**Abstract** Predicting adaptive phenotypic evolution depends on invariable selection gradients and on the stability of the genetic covariances between the component traits of the multivariate phenotype. We describe the evolution of six traits of locomotion behavior and body size in the nematode *Caenorhabditis elegans* for 50 generations of adaptation to a novel environment. We show that the direction of adaptive multivariate phenotypic evolution can be predicted from the ancestral selection differentials, particularly when the traits were measured in the new environment. Interestingly, the evolution of individual traits does not always occur in the direction of selection, nor are trait responses to selection always homogeneous among replicate populations. These observations are explained because the phenotypic dimension with most of the ancestral standing genetic variation only partially aligns with the phenotypic dimension under directional selection. These findings validate selection theory and suggest that the direction of multivariate adaptive phenotypic evolution is predictable for tens of generations.

## Editor's evaluation

This is an important paper that takes advantage of a comprehensive evolutionary genetic dataset to tease apart the relationship between genetic variation, selection, and phenotypic divergence over 50 generations. The evidence supporting the conclusions is robust and aligns with a growing body of work that shows patterns of variation can predict divergence over long periods of time and also that evolution does not always occur in the direction of selection, particularly when selection is acting on genetically correlated traits. The questions addressed in this study will particularly appeal to evolutionary biologists and quantitative geneticists.

**\*For correspondence:**
mallard@bio.ens.psl.eu (FM);
teotonio@bio.ens.psl.eu (HT)

**Competing interest:** The authors declare that no competing interests exist.

## Introduction

Predicting adaptive phenotypic evolution is an important research goal in evolutionary biology, agronomy, or in conservation policy (***Arnold, 2014***; ***Nosil et al., 2020***; ***Wortel et al., 2023***). It is generally accepted that predicting adaptive phenotypic evolution should be done in the context of the whole organism because organisms are not mere collections of genetically or environmentally independent traits (***Gould and Lewontin, 1979***; ***Wagner, 2001***). Many traits in natural populations are heritable across generations and under natural selection (***Walsh and Lynch, 2018***). Often, however, phenotypic evolution is not observed (***Merilä et al., 2001***; ***Pujol et al., 2018***), or is of opposite direction than predicted, because of environmental or genetic correlations between the traits of interest with unmeasured traits (***Morrissey et al., 2012***; ***Hajduk et al., 2020***).

Current approaches to predict phenotypic evolution during adaptation to novel environments, at least for infinite sexual populations and under infinitesimal assumptions of trait inheritance (***Barton et al., 2017***), rely on Simpson's adaptive landscape idea and its formalization by R. Lande: $\Delta \bar{z} = G\beta$ (***Simpson, 1944***; ***Lande, 1979***; ***Arnold et al., 2001***). In Lande's equation, the adaptive evolution of multiple traits' means

over one generation (the column vector $\Delta\bar{z}$) is modeled as a function of the $G$-matrix, which summarizes the genetic structure and heritable transmission of traits from parents to offspring, with the additive genetic variances of traits being represented in the diagonal entries and the additive genetic covariances between them in the off-diagonal entries. The direction and magnitude of phenotypic evolution depend on the size, due to genetic variances, and shape, due to genetic covariances, of the $G$-matrix. In particular, trait combinations with more genetic variation (henceforth, called canonical traits), allow for faster and more directed responses to selection (*Fisher, 1930*; *Lande, 1980*; *Schluter, 1996*; *Blows and McGuigan, 2015*). In Lande's equation, phenotypic evolution is also modeled as a function of the selection gradients on each trait (the vector $\beta$), gradients that describe the strength of directional selection on each trait in populations that can be under stabilizing/disruptive selection on each trait and under correlated selection between traits (*Lande and Arnold, 1983*). Correlated selection determines the shape of the selection surface: when traits with genetic variation are not aligned with the directional selection gradients, phenotypic evolution will be slower and distorted, resulting in less direct responses (*Phillips and Arnold, 1989*; *Arnold et al., 2001*; *Svensson et al., 2021*).

Lande's equation predicts adaptive phenotypic evolution but might fail when indirect selection is important. Indirect selection results from unmeasured traits being genetically correlated with the measured traits or when there is correlated selection between measured and unmeasured traits (*Lande and Arnold, 1983*; *Rausher, 1992*). Using Robertson's Secondary Theorem of Natural Selection (STNS) (*Robertson, 1968*; *Walsh and Lynch, 2018*), $\Delta\bar{z} = s = \sigma_g(z, w)$, trait changes over one episode of selection are accurately predicted because the selection differentials ($s$) equal the (additive) genetic covariance ($\sigma_g$) of the trait with relative fitness ($w$) and thus the trait's breeding value change in one generation, regardless of unmeasured traits (*Morrissey et al., 2010*; *Morrissey and Bonnet, 2019*). However, distinguishing direct from indirect selection is not possible with Robertson's STNS, which led *Stinchcombe et al., 2014* to propose combining its merits with those of Lande's equation in a single statistical framework to predict adaptive phenotypic evolution, see also *Etterson and Shaw, 2001*; *Morrissey et al., 2012*; *Hajduk et al., 2020*. Using Lande's equation retrospectively, one can estimate 'genetic' selection gradients as $\beta_g = G^{-1}s = G^{-1}(\bar{z}_a - \bar{z}_g)$, where the selection differentials are defined by the difference between the trait measured in an ancestral population ($\bar{z}_a$), and the trait of a diverging population ($\bar{z}_g$) as predicted by the STNS.

Unfortunately, using Lande's equation with the genetic selection gradients to predict adaptive phenotypic evolution over several generations depends not only on invariable selection gradients but also on the stability of the $G$-matrix. The input of genetic covariances by pleiotropic mutations could in the long-term of mutation-selection balance (time being scaled by the effective population size) be aligned with correlated selection and eventually explain phenotypic differentiation among populations and species (*Lande, 1980*; *Jones et al., 2007*; *Jones et al., 2014*; *Chebib and Guillaume, 2017*; *Houle et al., 2017*; *Farhadifar et al., 2015*; *Svensson et al., 2021*). However, many studies find more standing genetic variation in natural populations than expected at mutation-selection balance (*Walsh and Lynch, 2018*; *Sella and Barton, 2019*). In part, this is because in the initial stages of adaptation selection might not be weak relative to recombination as required by theory (*Lande, 1980*; *Nagylaki, 1992*; *Turelli and Barton, 1994*), in part this is because selection is not constant or uniform in temporally changing or spatially heterogeneous environments (*Gomulkiewicz and Houle, 2009*; *Chevin et al., 2010*; *de Villemereuil et al., 2020*; *Walter, 2023*). In addition, the $G$-matrix is bound to evolve in the short-term because of selection (*Cheverud, 1996*; *Barton and Turelli, 1987*; *Turelli, 1988*; *Shaw et al., 1995*), although there is mixed empirical evidence that the $G$-matrix can evolve to align with the orientations of selection (*Steppan et al., 2002*; *Arnold et al., 2008*; *Chenoweth et al., 2010*; *Ramakers et al., 2018*; *Johansson et al., 2021*). The reduction in the size of the $G$-matrix due to drift can be predicted because it is inversely proportional to the effective population size (*Lande, 1976*; *Lynch and Hill, 1986*; *Barton et al., 2017*), but the shape of the $G$-matrix will change unpredictably because all populations are finite, and bottlenecks and founder effects are common (*Phillips et al., 2001*; *Phillips and McGuigan, 2006*). Hence, besides selection, drift might also impact ongoing phenotypic evolution long before the mutation-selection balance is reached (*Whitlock et al., 2002*; *Mallard et al., 2022*).

Here, we ask if Lande's equation with the genetic selection gradients predicts the direction of adaptive phenotypic evolution for more than one generation. By adaptive phenotypic evolution we mean that multivariate trait responses to indirect or direct selection are correlated with adaptation to a novel and fixed target environment. We focus on the locomotion behavior and body size of the hermaphroditic nematode *Caenorhabditis elegans* in replicate populations gradually evolving to a high salt (NaCl) concentration in

their growth-media for 35 generations and then kept in the constant high salt environment for an extra 15 generations. Our ancestral population was domesticated to constant low salt conditions for 140 generations and contains abundant but neutral standing genetic variation for locomotion behavior (*Mallard et al., 2022*). Although replicate populations were maintained at high population sizes, our experimental regime exacerbates drift and inbreeding because of a slow rate of environmental change until reaching the target high salt environment (*Guzella et al., 2018*), and because high salt favors hermaphrodite self-fertilization (*Theologidis et al., 2014*).

Osmotic pressure from high salt concentration in the growth media shrinks individual body size because of water cell loss (*Urso and Lamitina, 2021*). For the ancestral population, and in our laboratory conditions, high salt also lowers embryo to adult survival and retards growth until maturity (*Theologidis et al., 2014*). As hermaphrodites cannot mate with each other, delayed male development results in hermaphrodites reproducing mostly by self-fertilization. During domestication, movement was reduced from that observed among wild isolates (*Mallard et al., 2022*), and hermaphrodites can further reduce movement in high salt during experimental evolution as males become less frequent and cannot harass them (*Barr et al., 2018*; *Cutter et al., 2019*). We further know that the canonical trait of locomotion behavior with the most mutational variance in low salt conditions differs from that with the most standing genetic variation found after domestication (*Mallard et al., 2023*). Consistent with these observations, several studies have shown that *C. elegans* mutants insensitive to high salt have specific defects in backward or forward movement, in some of these mutants independently of body size effects (*Fujiwara et al., 2002*; *Swierczek et al., 2011*; *Zhen and Samuel, 2015*). On the other hand, movement can increase during experimental evolution due to more foraging and dwelling, as the bacterial food is not as dense in high salt (*Gray et al., 2005*). Both foraging, dwelling and mate interactions in *C. elegans* can be described as a complex collection of distinct behavioral states, which vary in the duration of activity and movement direction (*Flavell et al., 2020*). All these considerations suggest that multiple traits in locomotion behavior can respond to selection but that it is difficult to a priori define which ones are genetically or environmentally independent. For this reason, we mathematically define individual locomotion behavior in 1-dimensional space by six traits, the six transition rates between stillness, moving forward, and moving backward (*Mallard et al., 2022*, *Mallard et al., 2023*). Body size is also analyzed as a seventh trait.

In what follows, we ask whether the ancestral phenotypic plasticity between high and low salt environments is aligned with the ancestral $G$-matrix. We use selection differentials on the seven traits in low salt and high salt environments to predict phenotypic evolution by describing the phenotypic and genetic divergence in high salt. Using Lande's retrospective equation, we ask if the genetic selection gradients measured in the ancestral population match the phenotypic selection gradients.

## Results

### Experimental design and analyses

The ancestral population for experimental evolution (A6140) was ultimately derived from a hybrid population of 16 isolates and domestication to a standard laboratory environment in low salt (25 mM NaCl) growth-media conditions for 140 generations (*Teotonio et al., 2012*; *Noble et al., 2017*). GA[1,2,4] replicate populations were derived from A6140, with limited founder effects, and independently exposed for 35 generations to a gradual change in salt concentration (8 mM increase each generation) and then kept in constant high salt (305 mM NaCl) for 15 generations. During the experiment, replicate populations were maintained at high population sizes (N=$10^4$), and from generation 35 onwards, hermaphrodite self-fertilization became predominant (*Theologidis et al., 2014*). Using genomic data, effective population sizes have been estimated at N$_e$=$10^3$ in the domestication low salt environment and under partial selfing (*Chelo and Teotónio, 2013*).

From the ancestral population (A6140), and the three replicate populations at generation 50 (GA[1,2,4]50), inbred lines were derived by self-fertilization of hermaphrodites (*Noble et al., 2017*; *Chelo et al., 2019*). Inbred lines were measured for hermaphrodite locomotion behavior and body size at the usual reproduction time of experimental evolution in low and high salt (186, 61, 61, and 42 lines from the ancestral and evolved populations, respectively, with most lines being phenotyped twice; see Methods for details). Six traits defined locomotion behavior: the transition rates between movement states, stillness, forward, and backward (*Mallard et al., 2022*). For the inbred lines of the

**Table 1.** Notation.

| Variable | Definition |
|---|---|
| $w$ | relative fitness in high salt, the self-fertility of hermaphrodites; from *Chelo et al., 2019* |
| $q_{i,j}$ | transition rates between the movement states $i$ and $j$; see *Equation 1* |
| $G$ | genetic (co)variance matrix of transition rates and body size; see *Equation 2* |
| $G_{qw}$ | genetic (co)variance matrix of transition rates, body size, and self-fertility |
| $S_k$ | ancestral selection differentials in high salt, with $k$ the salt environment where traits were measured; last column of $G_{qw}$ |
| $\beta_g$ | vector of genetic selection gradients; see *Equation 6* |
| $\beta$ | vector of phenotypic selection gradients; see *Equation 7* |
| SSCP | Sum-of-Squares and Cross-Product matrices for the environment and population factors; from MANOVA |
| $d_{max}$ | 1st eigenvector of the population factor SSCP-matrix in high salt |
| $\delta p$ | 1st eigenvector of the environment factor SSCP-matrix, for the ancestral population |
| $g_{max}$ | 1st eigenvector of the ancestral $G$-matrix, one for each salt environment |
| $e_{max}$ | first eigenvector of the random skewer $R$-matrix representing the main canonical trait differentiating the four $G$-matrices in high salt |
| $\Delta \bar{q}_k$ | Mean difference of the GA[1,2,4]50 populations from A6140, with $k$ the salt environment where traits were measured; from MANOVA |
| $\lambda_i$ | eigenvalue of the $i$th eigenvector |
| $\Theta$ | the angle between eigenvectors of ancestral genetic variation and $\delta p, d_{max}$, or $e_{max}$; see *Equation 3* |
| $\Pi$ | proportion of $G$-matrix variance along $\delta p$, $d_{max}$, or $e_{max}$; see *Equation 5* |

ancestral population, we also use self-fertility data at the usual reproduction time of experimental evolution in high salt, as previously reported by *Chelo et al., 2019*. Finally, we measure the extent of adaptation to high salt conditions using the outbred experimental populations from which the inbred lines were derived. Assays were designed so that grandmaternal and maternal environmental effects were the same for all the samples being compared.

With this data (*Figure 1—source data 1*, *Figure 1—source data 2*), we model phenotypic plasticity (mean population differences between environments) and standing genetic variation for locomotion behavior and body size in the ancestral population, the evolution of locomotion behavior and body size, and $G$-matrix evolution in the three replicate populations at generation 50. We estimated phenotypic plasticity and phenotypic differentiation in a multivariate MANOVA model and compared it with a univariate response model similarly defined (see Methods). The MANOVA allows us to test for ancestral phenotypic plasticity and phenotypic divergence while accounting for potentially correlated trait variation. The univariate approach allows us to estimate the inbred lines trait values and to test for the phenotypic divergence of each replicate population relative to the ancestral population but does not account for correlated variation in multivariate phenotypic space. Markov chain Monte Carlo methods were used in a Bayesian framework to estimate the $G$-matrix as half the among-line variance (see Methods) and, for the ancestral population, the $G$-matrix together with the genetic (co)variances between traits and fitness. *Table 1* defines the variables employed.

## Ancestral population
### Phenotypic plasticity between salt environments
Before experimental evolution to high salt, we started by characterizing phenotypic and genetic variation in low and high salt environments in the ancestral domesticated population. We find extensive

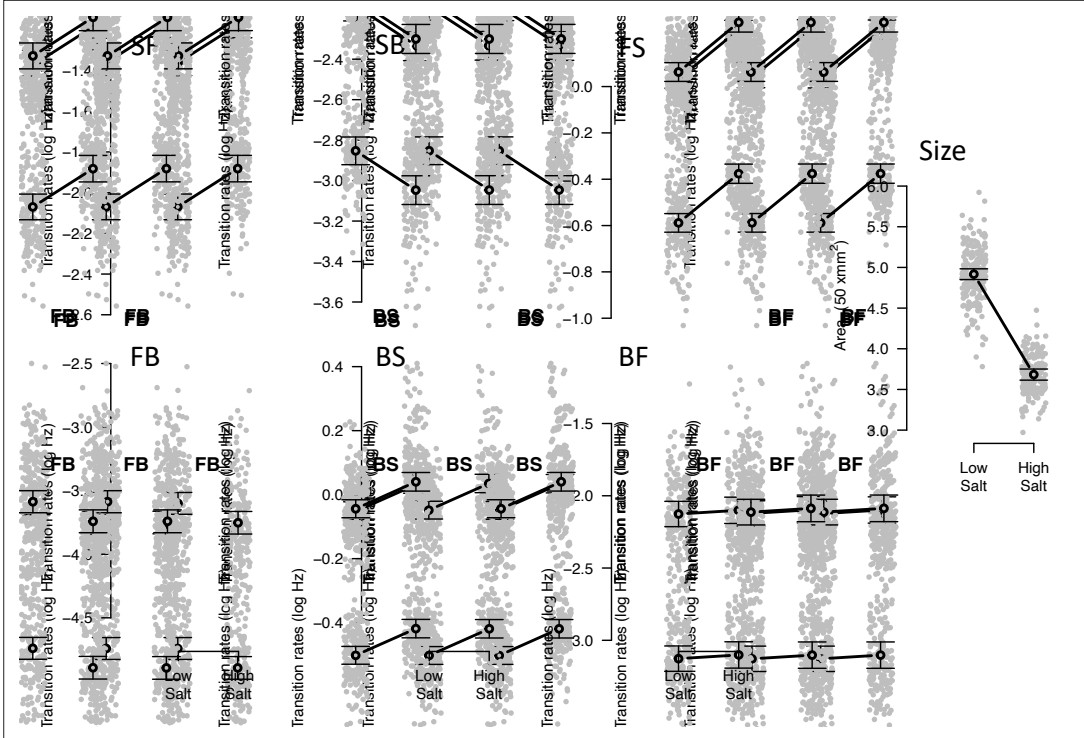

**Figure 1.** Phenotypic plasticity of the ancestral population. Gray dots indicate the trait values (BLUPs) estimated for each inbred line in the low and high salt environments: F for 'forward,' B for 'backward', and S for 'still,' left to right order indicating movement direction. Gray circles and bars indicate the mean 95% confidence intervals least-square estimates using the univariate approach (see Methods). Significant differences between environments are indicated with a line, when using the multivariate approach (**Table 2**). Figure source code is linked here - Multivariate analysis of variance (MANOVA) and figures/tables export scripts.

The online version of this article includes the following source data, source code, and figure supplement(s) for figure 1:

**Source data 1.** Raw data for analysis including all design and environmental covariates.

**Source data 2.** Sample sizes, see table.

**Source data 3.** Multivariate analysis of variance (MANOVA) results, see table.

**Source data 4.** Multivariate analysis of variance (MANOVA) results for the ancestral population by each trait, see table.

**Source data 5.** Eigendecomposition of the MANOVA SSCP matrix for the environment factor, see table.

**Figure supplement 1.** Multivariate and univariate models' environmental effects.

**Figure supplement 1—source code 1.** See Figure script.

**Figure supplement 1—source data 1.** Univariate models' contrasts. See table.

phenotypic plasticity for locomotion behavior traits and body size (**Figure 1**, **Table 2**). Because we employed univariate and multivariate approaches to model phenotypic plasticity (see Methods), we compared the estimated environmental effects using both approaches. We find that univariate and multivariate modeling approaches give similar results (**Figure 1—figure supplement 1**, **Figure 1— source data 3**, **Figure 1—source data 4**). Most transition rates are plastic with salt, except from forward to backward movement states. As expected, body size shrinks in high salt.

The multivariate approach (MANOVA) allows us to determine the phenotypic dimension of ancestral phenotypic plasticity ($\delta p$) that most responds to salt environmental variation (see Methods) as the first eigenvector of the MANOVA SSCP-matrix for the environment factor (**Figure 1—source data 5**). Transition rates from still to forward or to backward (SF or SB) have opposite loading signs in $\delta p$ (**Table 3**). Body size has the same sign of the transition rates from still and from forward to backward (SB and FB) and the opposite sign relative to the other transition rates.

**Table 2.** MANOVA results for ancestral phenotypic plasticity and phenotypic differentiation.

| Factor | Df | Wilks | approx.F | num.DF | den.Df | Prob(>F) |
|---|---|---|---|---|---|---|
| Environment | 1 | 0.137 | 523.4 | 7 | 583 | 2.20E-16 |
| Population | 3 | 0.310 | 40.2 | 21 | 1674.6 | 2.20E-16 |
| Environment:Population | 3 | 0.828 | 5.4 | 21 | 1674.6 | 3.99E-14 |
| Residuals | 589 | | | | | |

Notes: The Environment factor refers to the phenotypic difference between high and low salt environments for the ancestral population (**Figure 1**). The Population factor refers to the phenotypic differences between the four populations in the high salt environment (A6140 and GA[1,2,4]50); (**Figure 6**). The interaction between Environment and Population refers to the change in phenotypic difference between the four populations between the two environments, that is, to the evolution of phenotypic plasticity (**Figure 6—figure supplement 2**). The intercept in this MANOVA model is the ancestral population trait values in the high salt environment. Full model results, including the effects of assay design and environmental covariates (block, temperature, density, etc.), can be found in **Figure 1—source data 3**.

## Standing genetic variation

We next partitioned the phenotypic (co)variances among the inbred lines of the ancestor population into the genetic ($G$-matrix) and the residual error (co)variances of transition rates and body size (see Methods). Ancestral $G$-matrices were estimated separately by salt environment, assuming genome-wide homozygosity and no directional non-additive genetic effects. Our estimates are robust to changes in the prior distributions (**Figure 2—figure supplement 1**).

We find significant genetic variance for all transition rates and body size, in high and low salt environments, except for the transition rate from backward to still in low salt (**Figure 2**, **Figure 2—figure supplement 2**). The genetic covariances between transition rates and/or body size show similar values, albeit of lower magnitude in low salt (**Figure 2A**). The $G$-matrix size is similar between environments (**Figure 2B**). In both high and low salt environments, transition rates from still to forward or to

**Table 3.** Canonical traits of ancestral standing variation, divergence, and selection in high salt.

| Trait | $\delta p(1)$ | $g_{max}(1)$ | $g_2(1)$ | $g_3(1)$ | $d_{max}(1)$ | $e_{max}(1)$ | $\beta_g(2)$ | $m_{max}(3)$ | $m_2(3)$ |
|---|---|---|---|---|---|---|---|---|---|
| SF | 0.148 | –0.360 | –0.388 | 0.241 | –0.225 | –0.331 | –0.93 | –0.402 | –0.125 |
| SB | –0.103 | –0.459 | –0.409 | 0.394 | –0.365 | –0.502 | 0.93 | –0.224 | –0.209 |
| FS | 0.161 | 0.267 | 0.303 | 0.129 | 0.284 | 0.378 | 0.35 | 0.607 | 0.257 |
| FB | –0.039 | 0.532 | –0.502 | –0.117 | 0.517 | 0.459 | 0.32 | 0.632 | –0.518 |
| BS | 0.094 | 0.142 | 0.153 | 0.035 | 0.253 | 0.145 | –0.96 | 0.077 | 0.106 |
| BF | 0.062 | 0.467 | –0.496 | 0.150 | 0.473 | 0.498 | –0.82 | –0.096 | –0.765 |
| Size | –0.963 | –0.258 | –0.264 | –0.856 | –0.425 | –0.132 | 0.83 | 0.069 | –0.102 |
| | 99% | 76% | 9% | 6% | 95% | - | - | 39% | 37% |

Notes: (1) trait loadings of eigenvectors defined in **Table 1**, for the high salt environment; (2) modes of genetic selection gradients posterior distributions from **Figure 9**; (3) trait loadings of the first two eigenvectors of the mutational (co)variances matrix in low salt, re-analysis of locomotion behavior data with body size, from mutation accumulation lines reported in **Mallard et al., 2023** (see Discussion). The bottom row shows the percent variation each eigenvector explains, when relevant.

The online version of this article includes the following source data for table 3:

**Source data 1.** Eigendecomposition of environmental effects in the ancestral population, see table.

**Source data 2.** Eigendecomposition of the high salt $G$

-matrix, see table.

**Source data 3.** Eigendecomposition of phenotypic differentiation, see table.

**Source data 4.** Genetic selection gradients for traits measured in high salt, see table.

**Source data 5.** Eigendecomposition of the mutation variance-covariance matrix, see table.

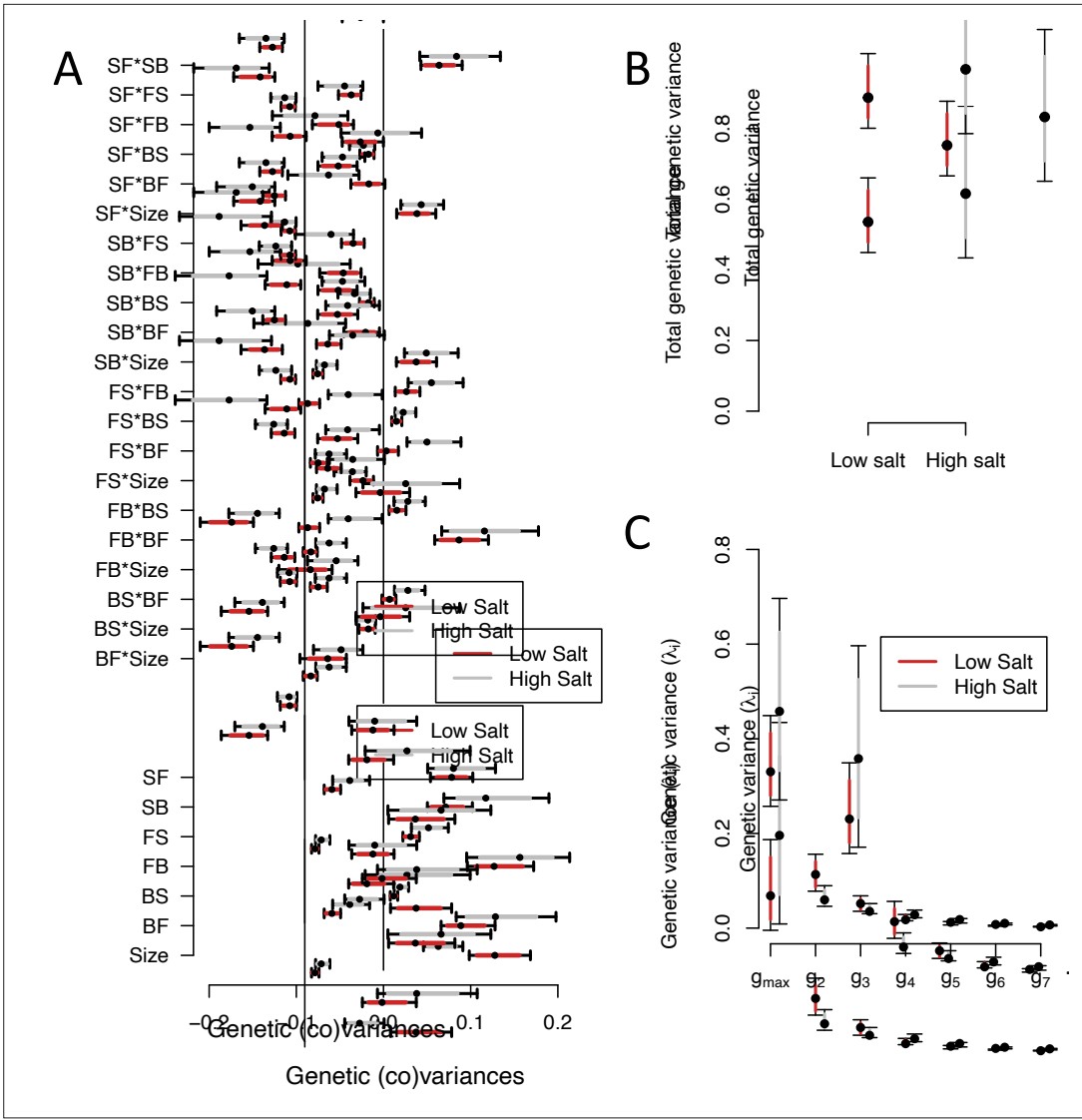

**Figure 2.** $G$-matrix of the ancestral population in low salt and high salt environments. (**A**). The bottom seven estimates indicate the genetic variances in transition rates and body size, top 15 estimates are the genetic covariances between the seven traits. (**B**). Total genetic variance in each environment is the trace of the $G$-matrices (**C**). Eigenvalues of the six eigenvectors for each $G$-matrix. For all panels, red (gray) indicates estimates in low (high) salt, with dots, and colored intervals the mode and the 83% or 95% credible intervals of the posterior distribution.

The online version of this article includes the following source data, source code, and figure supplement(s) for figure 2:

**Source code 1.** See G-matrix computation, *Figure 2* and table export scripts.

**Source data 1.** Ancestral $G$-matrix in low and high salt environments, see table.

**Source data 2.** Eigendecomposition of the ancestral $G$-matrices, see table.

**Figure supplement 1.** Varying priors for $G$-matrix estimation in the ancestral population.

**Figure supplement 1—source code 1.** See Figure script.

**Figure supplement 2.** Null distributions of genetic variances in the ancestral population.

**Figure supplement 3.** Eigendecomposition of null distributions of the ancestral $G$-matrix.

backward (SF or SB) are negatively correlated with the other transition rates and positively correlated with each other. Body size shows positive genetic covariances with SF and SB.

Eigendecomposition of the ancestral $G$-matrix in high or low salt reveals a similar structure between them (*Figure 2C*, *Figure 2—source data 2*). The first canonical trait ($g_{max}$, *Table 3*) encompasses most genetic

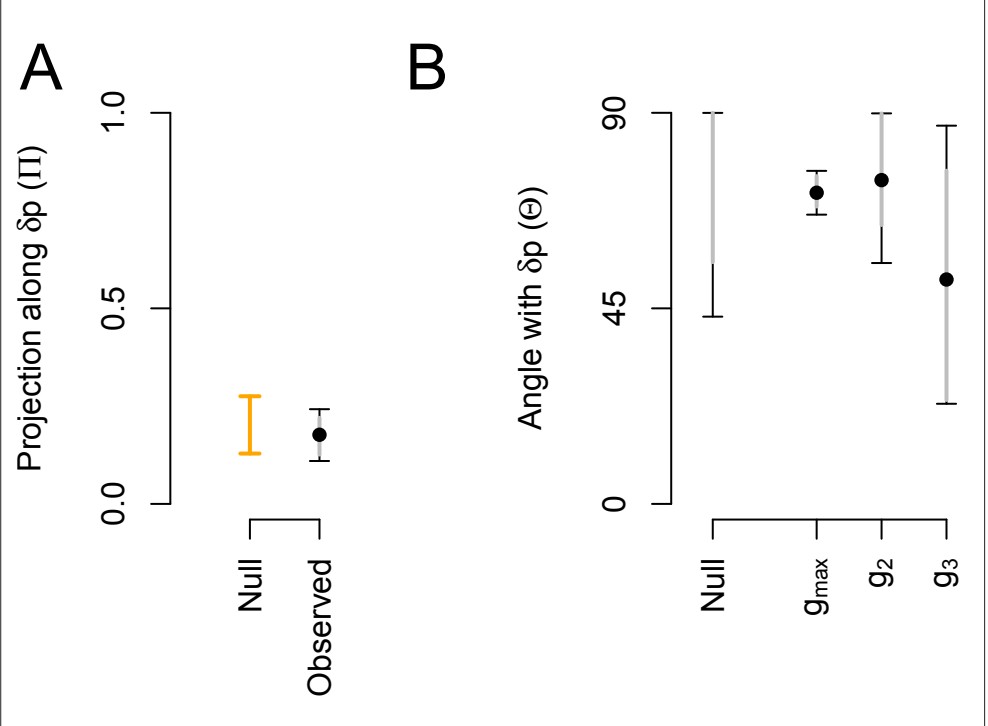

**Figure 3.** Aligment between phenotypic plasticity and standing genetic variation in high salt for the ancestral population. (**A**) Projection of the high salt $G$-matrix along the phenotypic plasticity canonical trait $\delta p$. Dots show the mean estimate with bars the 83% and 95% credible interval of the posterior $G$-matrix distribution. Orange bar shows the null 95% CI of the posterior distribution of modes of 1000 $G$-matrix randomized by inbred line and block identities (see Methods). (**B**). The angle ($\Theta$, **Equation 3**) between $\delta p$ and the first three eigenvectors of the ancestral $G$-matrix ($g_{max}$, $g_2$, and $g_3$). $\Theta$ does not differ from the random expectations. Dots show the mean estimate with bars the 83% and 95% credible interval of the posterior $G$-matrix distribution. The null expectation was obtained by computing the angle between pairs of random vectors sampled from a uniform distribution (see Methods).

The online version of this article includes the following source data for figure 3:

**Source code 1.** See **Figure 3** script.

**Source data 1.** Projections and angles (including CI) are shown in **Figure 3** as a table.

variation (75% in high salt, 63% in low salt). The next two canonical traits contain less genetic variation (between 6% and 20%) but are larger than null expectations (**Figure 2—figure supplement 3**). Only the first two canonical traits of the $G$-matrix have a similar trait loadings between environments (**Figure 2—source data 2**).

## Ancestral plasticity and genetic variation are not aligned

We compared the main canonical trait of phenotypic plasticity with the canonical traits of the high salt $G$-matrix. Phenotypic plasticity is not aligned with the $G$-matrix (**Figure 3**). This is because the amount of genetic variance along the dimension of phenotypic plasticity ($\delta p$) is not different than that expected by chance (**Figure 3A**), and also because the angle between $\delta p$ and $g_{max}$ is, if anything, larger than expected by chance (**Figure 3B**). $\delta p$ appears to similarly summarize environmental variation as the third canonical trait from the high salt $G$-matrix (**Table 3**), a trait that encompasses only 6% of standing genetic variation. As noted before, these differences stem from the association between still-to-forward and still-to-backward transition rates (SF and SB), which are genetically positive and environmentally negative, i.e., they have opposite signs in $g_{max}$ and $\delta p$, respectively. We suspect that positive associations between SF and SB reveal more dwelling, while negative association more individual foraging (**Flavell et al., 2020**).

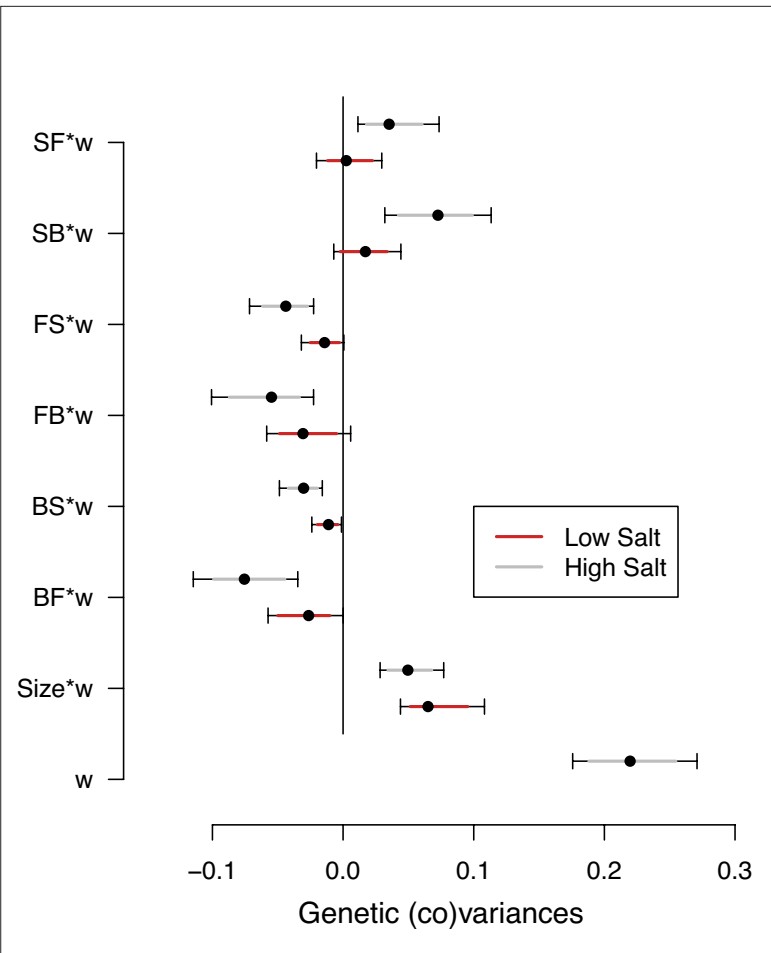

**Figure 4.** Selection differentials in the ancestral population. Ancestral genetic covariances between transition rates and body size measured in high salt (gray) or low salt (red) with high salt self-fertility. Dots and colored intervals show the mode and the 83% or 95% credible intervals of the posterior $G_{qw}$ distribution.

The online version of this article includes the following source data, source code, and figure supplement(s) for figure 4:

**Source code 1.** See $G_{qw}$ computation and *Figure 4* scripts.

**Source data 1.** Selection differentials' estimates, see table.

**Figure supplement 1.** Genetic (co)variances estimate from the $G$- and $G_{qw}$-matrices.

**Figure supplement 1—source code 1.** See Figure script.

**Figure supplement 2.** Self-fertility variation effects on selection differentials' estimates.

**Figure supplement 2—source code 1.** See Figure script.

## Selection differentials are similar across environments

Selection differentials on transition rates and body size measured in the two salt environments are their genetic covariances with the self-fertility measured in the high salt environment ($s_k$, *Table 1*). We used estimates of the ancestral $G_{qw}$-matrices to obtain these selection differentials (see Methods). $G$-matrices and $G_{qw}$-matrices estimates of genetic (co)variances in locomotion behavior and body size are similar (*Figure 4—figure supplement 1*). We find genetic variance for self-fertility in high salt (*Figure 4*, *Figure 4—source data 1*). We also find that the transition rates from still to forward or from still to backward (SF or SB) measured in high salt have positive genetic covariances with self-fertility, and all other transition rates have negative covariances (*Figure 4*). Small or no selection differentials exist in low salt transition rates, and only body size shows a clear positive selection differential in both environments. These results are robust to variation in self-fertility (*Figure 4—figure supplement 2*).

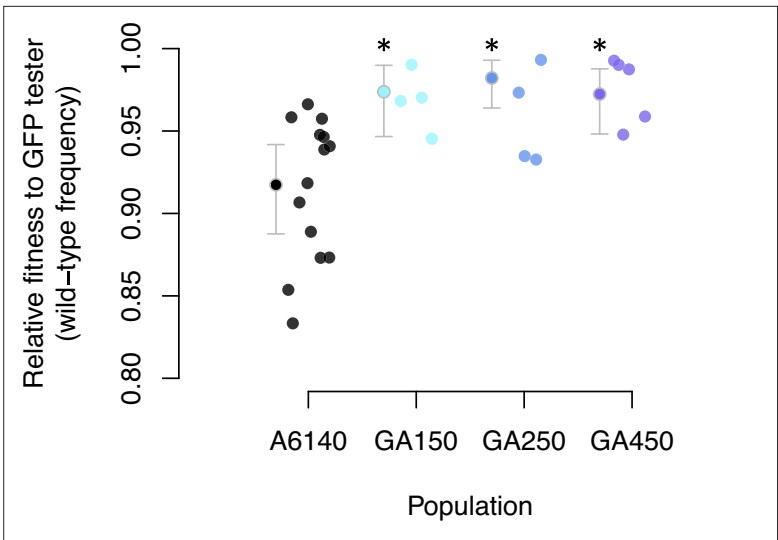

**Figure 5.** Adaptation to the high salt environment. Colored dots show the ratio of wild-type to green-fluorescent protein (GFP) alleles after one generation of pairwise competitions between the outbred experimental populations with a GFP-tester strain. Filled circles indicate the least-square mean estimates with 95% confidence intervals; asterisks indicate significant differences between each replicate population relative to the ancestral population.

The online version of this article includes the following source data for figure 5:

**Source code 1.** See data analysis and figure script.

**Source data 1.** Data for analysis, see table.

**Source data 2.** Population contrasts, see table.

## Evolutionary divergence

### Adaptation to the high salt environment

Having characterized ancestral standing variation, we describe the divergence of the three replicate populations after 50 generations of evolution. First, we measured the degree of adaptation to high salt by comparing the mean fitness of the GA[1,2,4]50 populations to the ancestral A6140 population in competition experiments against a tester GFP-strain (see Methods; *Figure 5—source data 1*). We find an increase in the mean fitness of all three replicate populations (*Figure 5*, *Figure 5—source data 2*). By generation 50, populations adapted to the high salt environment.

### Locomotion traits diverged in low and high salt environments

Concomitant with adaptation, there was phenotypic divergence for the locomotion traits and body size measured in high salt (*Figure 6*, *Table 2*). Estimates of phenotypic divergence are robust to multivariate and univariate modeling (*Figure 6—figure supplement 1*, *Figure 6—source data 1*). From the univariate models, we find that for each transition rate, at least one replicate GA population differed from the ancestor and that the three replicates showed significant divergence for three transition rates (*Figure 6*, *Figure 6—source data 2*). For body size we find that only one replicate populations diverged from the ancestral population. The amount of genetic variance did not limit phenotypic divergence, as the back-to-still and forward-to-still transition rates (BS and FS) diverged while showing the lowest genetic variances in the ancestral population (*Figure 2*). Eigendecomposition of the MANOVA SSCP-matrix for the population factor further reveals that a single canonical trait explains most phenotypic differentiation between the four populations ($d_{max}$; *Figure 6—source data 3*, *Table 3*).

In the low salt environment, there was less phenotypic divergence than in the high salt environment, with only three out of the six transition rates having at least two replicate populations significantly different from the ancestor (*Figure 6—figure supplement 2*). Unlike in high salt, body size in low salt showed a marked increase after experimental evolution in all replicate populations.

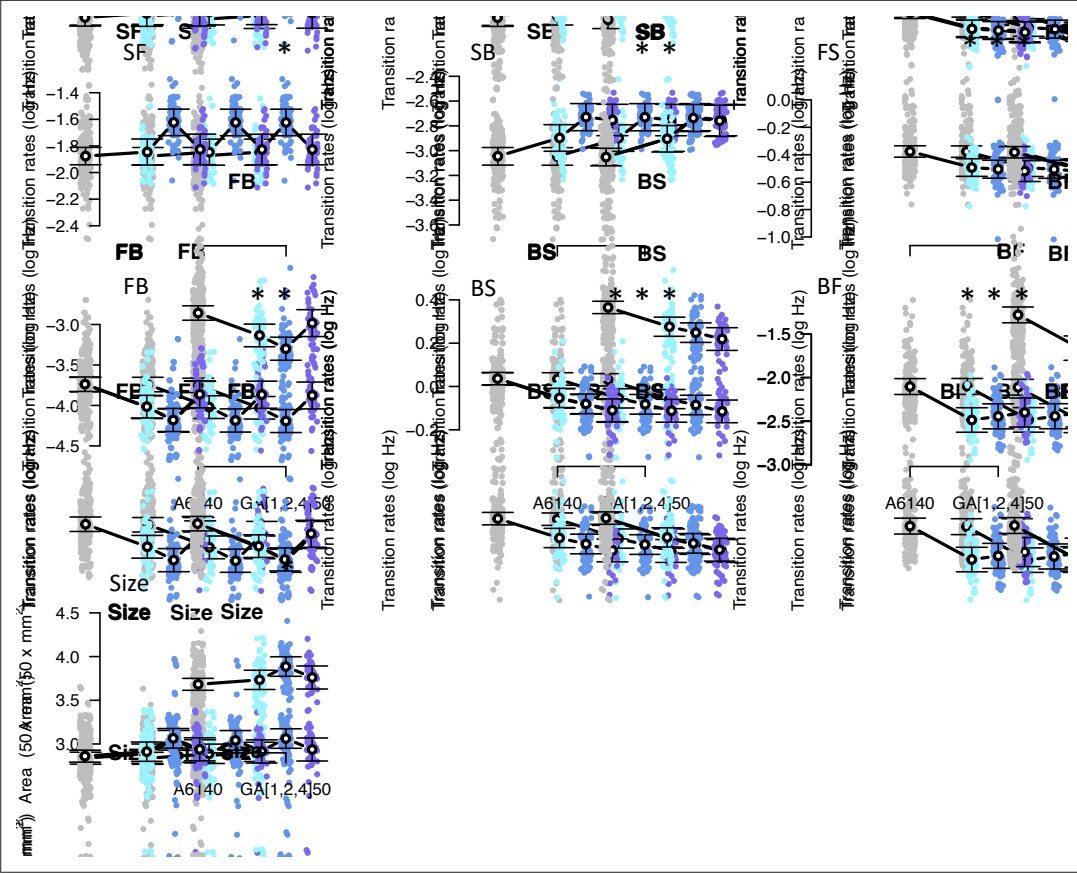

**Figure 6.** Phenotypic divergence in the high salt environment. Each panel shows the transition rates and body size as in *Figure 1*. Dots indicate the values estimated for each inbred line in a high salt environment, gray for the ancestral population, blues for the evolved replicate populations. Circles and bars indicate the mean and the 95% confidence intervals least-square estimates. Line shows significant differentiation between all four populations using the multivariate MANOVA approach (*Table 2*). Significant differences between each of the evolved populations and the ancestral population using the univariate approach are shown with asterisks (*Figure 6—source data 2*).

The online version of this article includes the following source data, source code, and figure supplement(s) for figure 6:

**Source code 1.** Multivariate analysis of variance (MANOVA) and figures/tables export scripts (as for *Figure 1*, also produces *Figure 6—figure supplement 2*).

**Source data 1.** Multivariate analysis of variance results, see table.

**Source data 2.** Contrasts between evolved and ancestral populations in high salt, see table.

**Source data 3.** Eigendecomposition of the MANOVA SSCP matrix of the phenotypic divergence, see table.

**Figure supplement 1.** Multivariate and univariate models' population estimates.

**Figure supplement 1—source code 1.** See Figure script.

**Figure supplement 2.** Phenotypic divergence in the low salt environment.

**Figure supplement 2—source data 1.** Contrasts between evolved and ancestral populations in low salt, see table.

## Genetic variance decreased during evolution

We next characterized genetic divergence by estimating the high salt $G$-matrices of the GA[1,2,4]50 populations and comparing them with the ancestral high salt $G$-matrix. We did not model the evolution of the $G$-matrix in the low salt environment. This analysis shows that the size of the high salt $G$-matrix was reduced during experimental evolution, independently of the replicate population (*Figure 7*, *Figure 7—source data 1*). However, we continue to find that most genetic variances for the

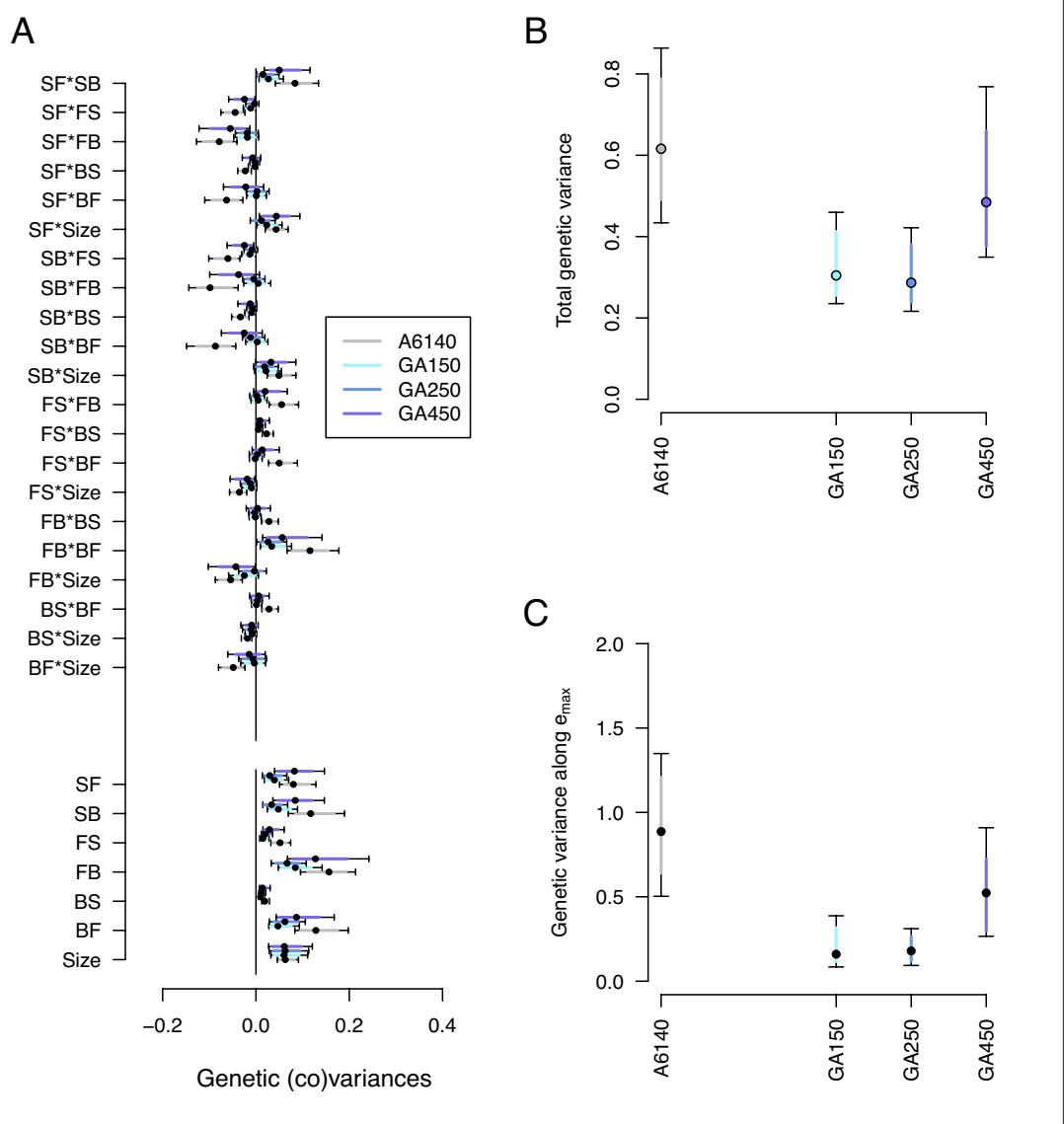

**Figure 7.** Genetic divergence in the high salt environment. (**A**) High salt $G$-matrix evolution of ancestral (gray) and evolved GA populations (blues). Eigendecomposition of the ancestral $G$-matrix (gray) can be found in ***Figure 2***, those of the evolved GA populations in ***Figure 7—figure supplement 2***. (**B**) Total $G$-matrix variance for each experimental population. (**C**) Genetic variance along $e_{max}$, the main canonical trait of genetic differentiation obtained after the random skewers analysis (see Methods, ***Table 1***). Dots and colored bars show the mode and the 83% or 95% credible intervals of the posterior distribution. Figure 7 sources linked here - matrix computation, random skewers analysis, and Figure 7 scripts. The Figure 7 scripts also produces all three figure supplements.

The online version of this article includes the following source data and figure supplement(s) for figure 7:

**Source data 1.** $G$-matrices of evolved populations in the high salt environment, see table.

**Figure supplement 1.** Null distributions of high salt genetic variances in the evolved populations.

**Figure supplement 1—source data 1.** Eigendecomposition of the evolved -matrices.

See table.

**Figure supplement 2.** Eigendecomposition of the high salt $G$-matrix of the evolved populations.

**Figure supplement 3.** Genetic variance of the high salt $G$-matrix in the evolved populations along the eigentraits of the ancestral population.

GA populations differ from null expectations (*Figure 7—figure supplement 1*). Eigendecomposition of the GA *G*-matrices indicates that 3–5 canonical traits differ from null expectations (*Figure 7—figure supplement 2*). Furthermore, evolved populations continued to have significant genetic variances in the three canonical traits of ancestral standing genetic variation (*Figure 7—figure supplement 3*).

Random skewers analysis shows that when projecting the four high salt *G*-matrices along 1000 random phenotypic directions, 250 of them showed a significant difference between the ancestral and at least one of the evolved populations (see Methods). Using these 250 vectors we build an *R*-matrix of genetic divergence between the four populations. Eigendecomposition of the *R*-matrix then revealed that a single canonical trait explains most divergence ($e_{max}$, *Table 1*). In this canonical trait at least 2 of the 3 evolved populations showed a reduction in variance (*Figure 7C*). An alternative eigentensor analysis to detect genetic divergence among the 4 populations confirms the random skewers analysis (see results in our GitHub appendix).

### Divergence along 'genetic lines of least resistance'

We asked whether phenotypic divergence ($d_{max}$) and genetic divergence ($e_{max}$) occurred along the dimensions of most ancestral genetic variation ($g_{max}$). For this analysis, we calculated the angle ($\Theta$, see *Equation 3* in Methods) and the proportion of overlap ($\Pi$, see *Equation 5*) between these canonical traits. *Table 3* summarizes the main canonical traits of ancestral standing variation, and of phenotypic and genetic divergence.

Most of the genetic variance of the ancestral high salt *G*-matrix along $d_{max}$ is higher than expected by chance (*Figure 8A*). This is because $d_{max}$ and $g_{max}$ are aligned and their angle is very small when compared with other canonical traits of ancestral standing genetic variation, or with a null expectations (*Figure 8B*). We do not find a significant proportion of genetic variance of the high salt *G*-matrix along $e_{max}$ (*Figure 8C*), but find a small angle between $e_{max}$ and $g_{max}$ (*Figure 8D*), which is indicative of a good alignment between these canonical traits.

## Indirect selection and predicting phenotypic evolution

### Expected and observed responses to selection are aligned in high salt

Using Lande's retrospective equation, we compared the genetic selection gradients obtained with selection differentials on traits measured in high salt ($\beta_g$; see Methods, *Equation 6*) to the phenotypic selection gradients obtained with the observed responses in high salt after 50 generations ($\beta$; see Methods, *equation 7*). The ancestral population's high salt *G*-matrix was assumed stable during experimental evolution. Credible intervals were obtained, however, by sampling the *G*-matrix from its posterior distribution, with fixed ancestral selection differentials ($s_k$; *Figure 4*), or fixed observed phenotypic divergence for each of the three replicate populations ($\Delta \bar{q}_k$; *Figure 6* for high salt, *Figure 6—figure supplement 2* for low salt).

We find that phenotypic selection gradients for all traits are highly heterogeneous because higher phenotypic divergence has an outsize effect on the mean and error estimates. Nonetheless, the phenotypic selection gradients overlap with the corresponding genetic selection gradients for at least one replicate population (*Figure 9A*; *Figure 9—source data 1*, *Figure 9—source data 2*). Evidence for a lack of overlap between selection gradients in two replicate populations is found for the transition rate backward-to-forward and for body size. Only for the transition rate SB is there an overlap of selection gradients for all three replicate populations.

Given replicate heterogeneity, testing whether selection theory predicts the direction of phenotypic evolution is possibly best estimated as the angle between expected and observed responses to selection. For traits measured in high salt, we find that the angle between selection differentials and observed phenotypic divergence is low for all replicates (*Figure 9B*). In contrast, when traits are measured in low salt, expected and observed responses are not aligned. Similarly, whether theory predicts the magnitude of phenotypic evolution can be estimated as the ratio between expected and observed responses. For all traits measured in high salt, and across replicate populations, observed phenotypic divergence is on average, across traits and replicates, 3.5 times the selection differentials (*Figure 9C*). For the traits measured in low salt, predictions about the magnitude of divergence further degenerate (not shown).

### Indirect versus direct selection

There is evidence of direct selection on SB and backward-to-forward (BF), as well as on body size (*Figure 9A*, *Figure 9—source data 1*). We find a positive genetic selection gradient for SB and a

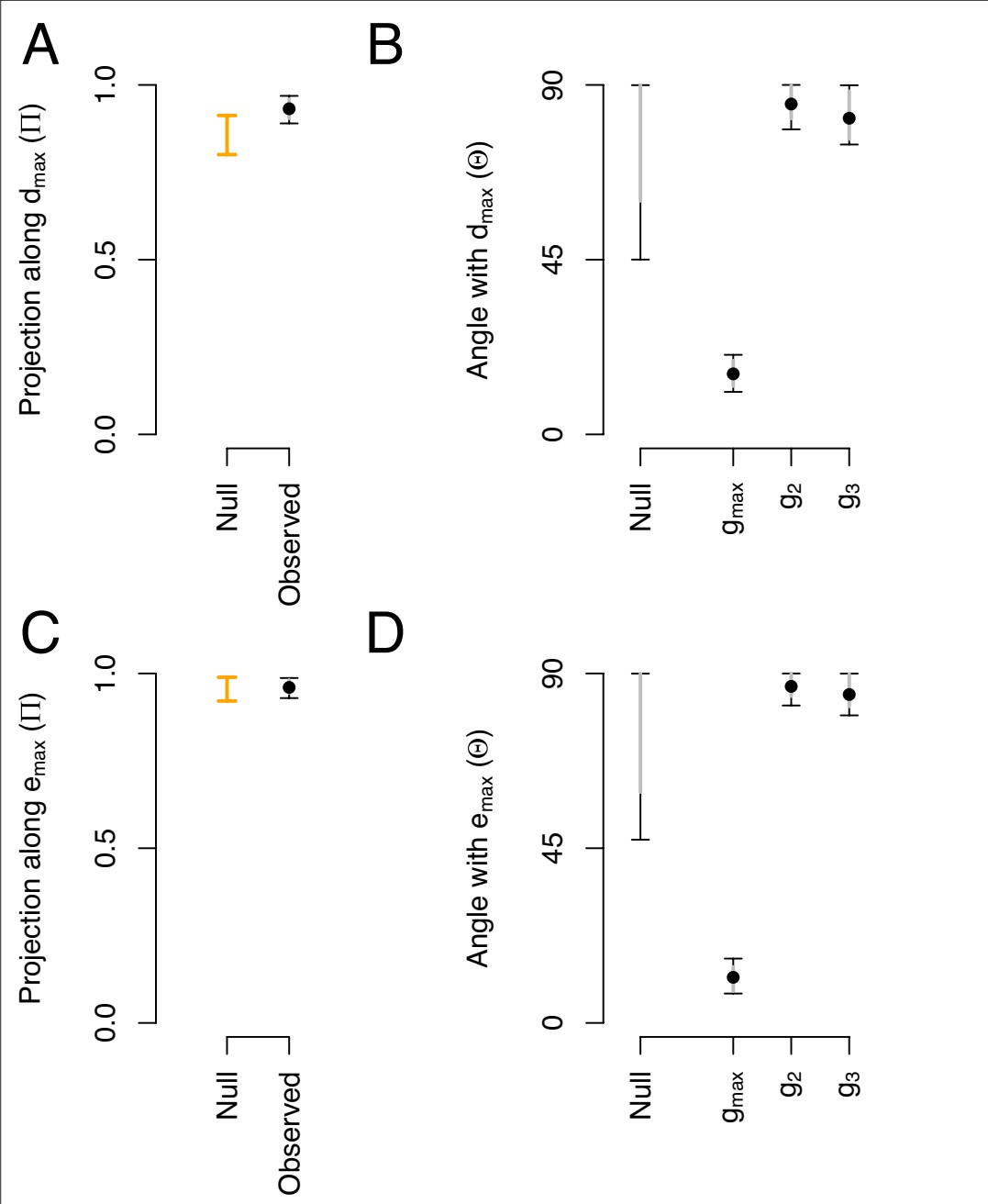

**Figure 8.** Phenotypic and genetic divergence alignments with ancestral standing variation. (**A**) Projection of the total ancestral genetic variance along the phenotypic divergence canonical trait $d_{max}$. Dots show the mean estimate with bars the 95% CI. Orange bar shows the null 95% CI after randomizing the $G$-matrix (see Methods). Mean of the observed posterior distribution (0.93) is outside the 95% CI of the randomized posterior modes (0.80–0.91). (**B**). The angle ($\Theta$) between $d_{max}$ and the first three eigenvectors of the ancestral $G$-matrix ($g_{max,2,3}$). The null expectation was obtained by computing the angle between 1000 pairs of random vectors. (**C and D**) Similar projection and angles as shown in (**A**) and (**B**) but with $e_{max}$ - the vector of the main genetic divergence - instead of $d_{max}$. In (**C**), the null and observed projections do not differ. Because $e_{max}$ and $g_{max}$ are almost aligned, both the observed and the null are very close to one (as $\Pi$ is estimated relatively to $\lambda_{max}$, see **Equation 5**) and the relative phenotypic variance between traits is conserved in the randomized $G$-matrices.

The online version of this article includes the following source data for figure 8:

**Source code 1.** See Figure AB and Figure CD scripts.

**Source data 1.** Projections and angles (including CI) shown in **Figure 8** as a table.

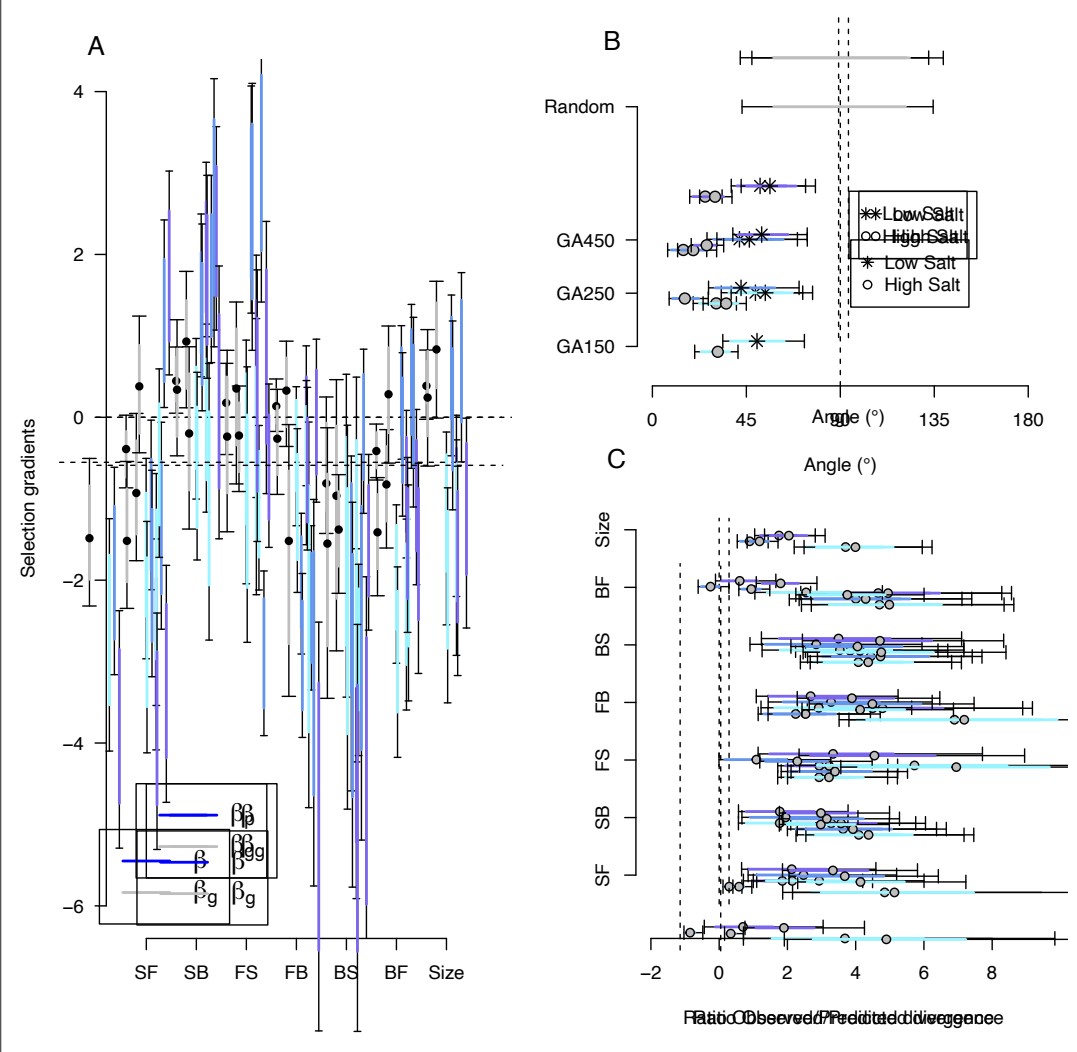

**Figure 9.** Predicting phenotypic evolution with Lande's equation. (**A**) Indirect and direct selection. Genetic selection gradients $\beta_g$ (gray) and phenotypic selection gradients $\beta$ (blues) for each replicate population, see *Equation 6* and *Equation 7*, respectively. $\beta$ were divided by 3.5 for scaling (the average ratio observed/predicted divergence, panel C) rather than by 140 (the total number of generations in the experiment) for visual convenience. (**B**) The direction of phenotypic evolution. Angle between the expected phenotypic divergence (selection differentials, $s_k$; *Figure 4*) and the observed phenotypic divergence at each replicate ($\Delta\bar{q}_k$; *Figure 6*). Circles show the results in the high salt environment and stars in the low salt environment. The expected angle by chance is in gray and was generated by computing 1000 angles between pairs of randomly generated vectors from a uniform distribution $\mathcal{U}^7(-1, 1)$. (**C**) The magnitude of phenotypic evolution. The ratio phenotypic divergence at each replicate ($\Delta\bar{q}_k$) with expected divergence ($s_k$). For all panels, dots/circles/stars and colored bars show the mode and the 83% or 95% credible intervals of the posterior distributions obtained by sampling in posterior distribution of the ancestral high salt $G$-matrix (*Figure 2*).

The online version of this article includes the following source data, source code, and figure supplement(s) for figure 9:

**Source code 1.** See *Figure 9* script that includes *Figure 9—figure supplement 1* and *Figure 9—figure supplement 2*.

**Source data 1.** Genetic selection gradients, see table.

**Source data 2.** Phenotypic selection gradients, see table.

**Figure supplement 1.** Genetic selection gradients bias due to low variance traits.

**Figure supplement 2.** Genetic selection gradients with a sampling of $G$-matrix and differentials.

**Figure supplement 2—source data 1.** Genetic selection gradients, see table.

negative genetic selection gradient for BF. There is also a positive selection gradient for body size, opposite in sign to its association with other traits in the canonical traits of standing genetic variation, and in the canonical traits of phenotypic and genetic divergence (*Table 3*). Traits with low genetic variance, FS and BS (*Figure 2—figure supplement 2*), do not appear to bias the genetic selection gradient estimates (*Figure 9—figure supplement 1*). However, none of the genetic selection gradient estimates differ from zero when sampling both the $G$-matrix and the selection differentials from their respective posterior distributions to obtain credible intervals (*Figure 9—figure supplement 2*).

## Discussion

Modeling approaches for predicting adaptive phenotypic evolution before mutation-selection balance is reached are based on Lande's equation (see Introduction). In Lande's equation, the trait change over one generation equals the ancestral $G$-matrix times the directional phenotypic selection gradients, but might not be accurate in the presence of indirect selection. This can be remediated by replacing the phenotypic selection gradients with the genetic selection gradients obtained after measuring selection differentials in the ancestral population. For predicting phenotypic evolution over several generations, however, one must assume invariable selection gradients and that the $G$-matrix is stable despite selection and drift. We sought to test the selection theory by finding whether we could predict phenotypic evolution for 50 generations of experimental evolution.

We followed seven traits with different environmental and genetic dependencies, the six transition rates between movement states and body size (*Table 3*). Our ancestral population was adapted to the low salt conditions (*Chelo and Teotónio, 2013*; *Theologidis et al., 2014*), before challenging three replicate populations to a gradual increase in the salt concentration in the growth media for 35 generations and 15 extra generations in high salt. Fifty generations of experimental evolution led to adaptation (*Figure 5*) and to phenotypic and genetic divergence (*Figure 6*, *Figure 7*). Body size measured in high salt did not consistently evolve among replicate populations but individual movement increased. This is because the transition rates from the still state have increased while those to the still state have decreased. Adaptive phenotypic divergence followed the direction of the canonical trait with more ancestral standing genetic variation (*Figure 8*, *Table 3*), and therefore, we could predict phenotypic evolution, though only its direction and when the component traits of the multivariate phenotype were measured in the high salt environment (*Figure 9*). When considering the component traits of the multivariate phenotype individually, and due to replicate population heterogeneity, we could confidently predict the evolution of only one of the seven traits followed (SB). We could not predict the magnitude of evolution for any individual trait. These findings are relatively unique because we described $G$-matrix evolution and measured the ancestral selection differentials to predict phenotypic evolution, but they add to the results of a growing number of experimental studies testing Lande's equation across tens of generations. For example, a recent re-analysis of up to 60 generations in constant and homogeneous environments, for five wing traits in *Drosophila melanogaster*, showed an alignment between the main canonical trait of genetic variation in the evolved populations with adaptive phenotypic divergence (*Yeaman et al., 2010*; *Walter, 2023*).

As the canonical trait explaining most variation in the ancestral population, $g_{max}$, also the second and third canonical traits differ from null expectations ($g_{2,3}$, *Figure 2*, *Figure 2—figure supplement 3*), and remain so after evolution (*Figure 7—figure supplement 3*), despite potential variance inflation problems due to the MCMC methods we employed (*Morrissey et al., 2012*; *Sztepanacz and Blows, 2017b*). For these ancestral canonical traits, selection must have been responsible for the observed loss of genetic variance, particularly after generation 35 (*Figure 7*). Assuming an infinitesimal model of trait inheritance, drift is expected to lead to a loss of genetic variance by ($1-1/2N_e$) at each generation (*Barton et al., 2017*). Even in the unrealistic situation of complete selfing during the experiment, and considering effective population sizes on the order of 1000 (*Chelo and Teotónio, 2013*), less than 5% was expected to be lost by genetic drift by generation 50, values that were not observed (only about half of the genetic variance was lost, *Figure 7B*). Supporting loss of variation by selection we before showed that allelic diversity at neutral single-nucleotide polymorphisms is reduced relative to ancestral levels by 5% by generation 22 (*Theologidis et al., 2014*), and to 20% only by generation 50 (*Chelo et al., 2019*). We did not, however, test whether the direction of phenotypic divergence occurred along the ancestral $g_{2,3}$ traits because they together explain 15% of the variation and there is a poor statistical power to do so. Furthermore, our previous work suggests that variation in the ancestral $g_{2,3}$

might have been lost by drift during the first 35 generations of the experiment such that they were perhaps of little consequence later on when populations reached the high salt environment (*Guzella et al., 2018*). Few studies have demonstrated that canonical traits of little genetic variance can influence selection responses (*Kirkpatrick, 2009*; *Blows and McGuigan, 2015*; *Sztepanacz and Blows, 2017a*). In one of the few examples, *Hine et al., 2014* showed that low-variance canonical traits of eight cuticular hydrocarbons in *Drosophila serrata* respond to artificial selection during six generations, though inconsistently among replicate populations. In our experiment, we suspect that showing that canonical traits with a small amount of genetic variation impact adaptive phenotypic divergence will require finding the quantitative trait loci (QTL) responsible for their expression (*Svensson et al., 2021*; *Kelly, 2009*). If these low-variance canonical traits influence adaptation then allele frequency dynamics at the relevant QTL because of selection might be detected when comparing genomic data between ancestral and derived populations (*Long et al., 2015*; *Barghi et al., 2020*).

Our findings also highlight the relationship between phenotypic plasticity and adaptation to novel environments (*Ghalambor et al., 2007*; *Pfennig et al., 2010*; *Teotónio et al., 2009*; *Draghi and Whitlock, 2012*; *Noble et al., 2019*). While the discussion has been on showing that population persistence is more likely if plasticity is aligned with the direction of selection (*Price et al., 2003*; *Lande, 2009*; *Chevin et al., 2010*), our results show that plasticity only reveals the topography of the adaptive landscape. In high salt conditions, populations move away from the ancestral phenotypic optimum (*Figure 1*), with an associated fitness cost (*Theologidis et al., 2014*). Adaptation to the high salt target environment after generation 35 presumably involved recovering a phenotype similar to that of the ancestral population that alleviated this fitness cost (*Table 3*). In particular, high salt in the ancestral population reduces body size and SB transition rates while increasing SF transition rates. Symmetrically, selection favors increased body size, increased SB, and decreased SF. However, because all three traits show positive genetic covariances with each other (*Figure 2*), even if plasticity is oriented with selection (but of the opposite sign), phenotypic evolution is constrained by a lack of genetic variation in the appropriate canonical trait. The ancestral population had genetic variation in the direction of selection (the canonical trait $g_3$, *Table 3*), but as argued above it was probably lost during gradual salt evolution because of drift such that when reaching high salt populations could not have further responded to selection (*Matuszewski et al., 2015*; *Guzella et al., 2018*). Future evolution in the direction of the ancestral multivariate phenotypic optimum, or close to it, should then be conditional on the appearance of de novo pleiotropic mutations. Assuming that mutational covariances do not vary with the environment, it is unclear that there can be much further phenotypic evolution as elsewhere we characterized mutational covariances and did not find any in the direction of selection (see *Table 3* and *Mallard et al., 2023*).

Ancestral phenotypic plasticity can thus be considered 'non-adaptive' (*Ghalambor et al., 2007*). Hence, it is unsurprising that we could not predict low salt phenotypic evolution. This is not explained because of a general lack of genetic variance in locomotion traits and body size in the ancestral population (*Figure 2*) but because their selection differentials were small or did not differ from zero (*Figure 4*). An exception is body size. Body size is reduced by high salt conditions (*Figure 1*), and there is probably direct selection for increased body size in high salt (*Figure 9*). However, there was little body size evolution in the high salt environment (*Figure 6*). In contrast, there was evolution of increased body size when it was measured in the low salt environment, as expected from the positive selection differentials in the ancestor population (*Figure 4*). Body size measured in low salt could only have evolved because of indirect selection by being genetically correlated with high salt body size or some other traits expressed in high salt. Hence, the evolution of plasticity in body size might be predictable although the evolution of plasticity in the multivariate phenotype is not. The fact that body size measured in low salt evolved further reveals that the evolved populations were in a region of the adaptive landscape that was not readily accessible to the ancestral population even if it had been domesticated to low salt conditions for 140 generations. In other words, body size evolution in low salt supports the long-held hypothesis that phenotypic plasticity facilitated the evolution of novelty (*Wagner and Lynch, 2010*; *Moczek et al., 2011*; *Levis et al., 2018*).

Robertson's covariance [$\sigma_g(z, w)$] was originally found to describe an episode of selection (the selection differentials, $s$) and later to predict adaptive phenotypic evolution over one generation ($\Delta\bar{z}$) as the secondary theorem of natural selection (*Walsh and Lynch, 2018*; *Hajduk et al., 2020*). Applications so far have been mostly limited to explaining the evolution of individual traits and when time-series of trait and

fitness are concurrently obtained with pedigrees such that estimates of trait breeding values and genetic selection gradients can be updated every few generations (*Stinchcombe et al., 2014*; *Morrissey et al., 2012*; *Hajduk et al., 2020*; *Hadfield, 2010*). One problem that has received particular attention has been to ask whether indirect selection or phenotypic plasticity and robustness explain phenotypic stasis, despite trait heritability and a significant phenotypic selection gradient on the observed trait (*Merilä et al., 2001*; *Kruuk et al., 2002*). For example, *Biquet et al., 2022* followed a blue tit population for more than 40 years, in which egg laying has changed to earlier spring dates, presumably because of climate change. Despite significant heritability of egg laying and directional selection for earlier dates, modeling the breeding values did not reveal any temporal trend, consistent with a lack of genetic covariance between egg laying date and fitness. There was thus no genetic divergence for laying date, which *Biquet et al., 2022* could attribute to phenotypic robustness and the stochastic nature of individual development to maturity. Conversely, using a similar approach, *Bonnet et al., 2017* found that evolution towards earlier parturition dates in a red deer population could be predicted and was consistent with the estimated change in breeding values. In a study following two traits, selection due to human harvesting of a prey species of wild salmon (for feeding domestic salmon) explained divergence in early maturity and small body sizes, despite directional selection for increased body size at maturity because of fishing (*Czorlich et al., 2022*). Our results suggest that when pedigrees are difficult to obtain, predicting the direction of adaptive multivariate phenotypic evolution for tens of generations may be possible without updating estimates of selection gradients and the $G$-matrix every few generations. This is because although the environment changed for 35 generations in our experiment directional selection was maintained and only the size of the $G$-matrix was reduced.

In sum, we have shown that using Lande's equation with genetic selection gradients is valid to predict the direction of phenotypic evolution in a new environment, after a gradual environmental change for 35 generations and 15 generations in the new environment. However, selection theory not necessarily predict the direction or magnitude of evolutionary change in all the component traits of the multivariate phenotype, especially if the traits are not measured in the new environment. This is because there are variable and complex genetic and environmental dependencies between individual traits. There are few experimental tests of selection theory such as ours. Therefore, more will be needed to generalize our results to natural populations, particularly those challenged by changing and heterogeneous environments.

## Materials and methods

### Experimental populations and environmental conditions

The ancestral population is named A6140, where 'A' stands for androdioecious, '6' for replicate six, and '140' for the number of generations of domestication to a standard laboratory environment (*Teotónio et al., 2017*). A6140 resulted from the hybridization of 16 founder wild strains during 33 generations followed by 140 generations characterized by 4 day discrete and non-overlapping life-cycles at N=10$^4$ census sizes and $N_e$=10$^3$ effective population sizes (*Teotonio et al., 2012*; *Chelo and Teotónio, 2013*; *Theologidis et al., 2014*). Our standard laboratory environment involves populations being maintained in 10 × 9 cm Petri dishes NGM-lite agar media containing 25 mM NaCl and a homogenous lawn of *E. coli* HT115 that served as food from the L1 larval stage until reproduction. Each Petri dish contains 1000 individuals which are mixed during reproduction, with embryos being collected and synchronized at the L1 larval stage to start a new generation.

We report the evolution of locomotion behavior and body size in three independent replicate populations (named GA[1,2,4] populations: 'G' for gradual, 'A' for androdioecious, '#' for replicate number). They were derived from splitting into three a single pool of at least 10$^4$ individuals sampled from the A6140 population. GA populations were maintained in the same conditions as during domestication except that the NGM-lite media was supplemented with 8 mM of NaCl at each generation for 35 generations and then kept at constant 305 mM NaCl for an additional 15 generations. Details about the derivation of the GA populations can be found in *Theologidis et al., 2014*. We refer to the NGM-lite 305 mM NaCl environment as the 'high salt' target environment, while the domestication 25 mM NaCl environment as the 'low salt' environment.

*C. elegans* is an androdioecious nematode, where hermaphrodites can self but outcross only when mated with males. Natural populations are depauperate of genetic diversity and males are rare due to a long history of selfing, selective sweeps, and background selection (*Andersen et al., 2012*;

*Rockman et al., 2010*). Under the domestication environment, however, outcrossing is readily maintained at frequencies between 60% and 100% (*Teotonio et al., 2012*, *Mallard et al., 2022*). In GA populations outcrossing is maintained at close to 100% for 35 generations, reduced to about 30% in GA1, 14% in GA2, and 5% in GA4, by generation 50 of the experiment (*Theologidis et al., 2014*).

As reported before, we derived inbred lines by selfing single hermaphrodites from the ancestor (A6140) and the three replicate populations at generation 50 (GA[1,2,4]50) for a minimum of 10 generations (*Noble et al., 2017*; *Noble et al., 2021*). Male frequency in the inbred lines is low, on the order of the mutation rate for the non-disjunction of the X-chromosome (*Teotónio et al., 2006*) – sex-determination is chromosomal with hermaphrodites XX and males XØ –.

Populations and inbred lines were cryogenically stored (*Stiernagle, 1999*), allowing for contemporaneous measurements of ancestral and evolved outbred populations and their inbred lines. Grandmaternal and maternal environmental effects are common to the samples being measured (*Teotónio et al., 2017*).

## Adaptation to high salt

We measured the increase in mean relative fitness among the ancestral population (A6140) and evolved populations at generation 50 (GA[1,2,4]50) using pairwise competition experiments between them and a tester line (*Teotónio et al., 2017*). As a tester, we employed an inbred line (EEV1402) derived by selfing from the A6140 population, and that expressed a green-fluorescent-protein (GFP) morphological marker (*Chelo et al., 2013*). For the assays, we revived the four populations and the tester line (>1000 individuals each) and let individuals reproduce and starve for 10 days. Starved individuals were then seeded on fresh plates with food at a density of 1000 L1 larvae in low salt. We grew them for two complete generations in high salt, except the GFP tester which was only grown in high salt for one generation. At the third generation, we seeded 500 L1 larvae of the GFP tester line together with 500 L1 larvae of 1 of 4 experimental populations in high salt. For A6140, we seeded 15 plates (technical replicates), for GA150 4 plates, for GA250 four plates, and for GA450 five plates. In each of these plates, 72 hr after L1 seeding, individuals were subject to the 'bleach/hatch-off' protocol, the standard of our life-cycle, to recover live embryos and, 24 hr later, synchronized L1 larvae. We scored an average of 169 larvae for GFP expression in each technical replicate.

The relative proportion of non-GFP to GFP measures the relative fitness of the experimental populations to the tester after one generation of competition (*Teotónio et al., 2017*). To analyze this data, we used a generalized linear model in R (*R Development Core Team, 2018*), testing for the evolution of the ratio non-GFP/GFP, assuming a binomial error distribution ('quasibinomial' family option) and allowing for overdispersion of the data. Post-hoc pairwise comparisons were performed between the ancestral and the evolved populations with Tukey tests using the *glht* function in the *multcomp* package in R (*Hothorn et al., 2008*).

## Locomotion behavior

Inbred lines were thawed from frozen stocks on 9 cm Petri plates and grown until exhaustion of food. This occurred 2–3 generations after thawing, after which individuals were washed, adults removed by centrifugation, and three plates per line seeded with 1000 larvae at mixed larval stages. Samples were then maintained in the standard domestication environment for two complete generations. At the assay generation (generation 4–6 generations post-thaw), starvation-synchronized L1 larvae were seeded in low and high salt. Adults were phenotyped for locomotion behavior 72 hr later at their usual reproduction time in one 9 cm plate (technical replicate). At the beginning of each assay we measured ambient temperature (T) and humidity (H) in the imaging room.

Given the number of lines to phenotype, we repeated the above protocol several times over several years, with each repetition defining a statistical 'block' on a given day. In total, we phenotyped 186 lines from the A6140 population and 61, 61, and 42 lines from each of the GA[1,2,4] populations, respectively, with most lines being phenotyped twice and always in separate blocks (average of 1.9 in low salt, and of 2 in high salt).

We imaged adult hermaphrodites using the Multi-Worm Tracker [version 1.3.0; *Swierczek et al., 2011* and used the materials and protocols of *Mallard et al., 2022*]. Each movie contains about 1000 tracks of hermaphrodites (called objects) with a mean duration of about 1 min. Standardized to a common frame rate (4 Hz), we filtered and extracted the number and persistence of tracked objects per movie and assigned movement states across consecutive frames as forward, still or backward

(assuming forward as the dominant direction of movement). Mean object density (D) per movie was also retrieved to be used as a covariate in modeling.

Locomotion behavior in 1-dimensional space is described by the transition rates between still (S), forward (F) and backward (B), plus the self-transition rates. Modeling is detailed in *Mallard et al., 2022*. Transition rates between movement states are assumed to follow a continuous time Markov process. The Markov process is a stochastic process modeling changes in movement state as a matrix Q. In our data, the Markovian memoryless assumption is only marginally violated (*Mallard et al., 2022*). The elements in Q, noted $q_{i,j}$, are the transition rates from state $i$ to state $j$ (off-diagonal elements for $i \neq j$, and with $q_{i,j} > 0$). This definition constrains self-transition rates (diagonal elements) to be of the opposite sign to the sum of the two transition rates leaving the relevant movement state:

$$q_{i,i} = -\sum_{j \neq i} q_{i,j} \tag{1}$$

This ensures that the probability of leaving a given movement state towards any other state during a waiting time $\Delta t$ is one minus the probability of remaining in the same state (see *Mallard et al., 2022* for a more detailed explanation). Therefore, only the six transition rates between movement states are mathematically independent and we thus ignore self-transition rates.

For estimation, we used log-likelihood models as defined in *Mallard et al., 2022* and specified them with the *msm* package (*Jackson, 2011*) in RStan [*Stan Development Team, 2018*, R version 3.3.2, RStan version 2.15.1]. Because $q_{i,j} > 0$, all analyses were performed on the natural log scale to ensure normality. We used multi-log normal prior distributions with the mean transition rate and a coefficient of variation $\ln(q_{i,j}) \sim \mathcal{N}(\ln(2), 0.6)$. We retained the means of the posterior distributions as the per-plate transition rates for all the subsequent analyses.

## Body size

We included the measurements of body size obtained from the Multi-Worm Tracker movies as a seventh trait. Movie frames were sampled only for forward tracked-objects to minimize posture variation. We then extracted the per-track object mean area (*Swierczek et al., 2011*). These values were summarized as the per-plate median of all track mean values. We then re-scaled these measurements so that the averaged phenotypic variance in each environment is roughly similar to the average transition rates phenotypic variance. This was done by multiplying the body size by a factor of 50. We chose this procedure rather than dividing all the phenotypic values by their mean, cf. *Houle et al., 2011*, because our transition rates' means are close to zero while spanning both negative and positive values. Dividing these transition rates by their means would lead to an artificial increase in phenotypic variance.

## Self-fertility

To estimate selection differentials, we used previously-published data on hermaphrodite self-fertility of the A6140 inbred lines in high salt (*Chelo et al., 2019*). Self-fertility was measured under environmental conditions that closely followed those of experimental evolution. An average of 42 hermaphrodites were scored for self-fertility per inbred line (minimum 22 and maximum 85 individuals). Self-fertility includes the fecundity of hermaphrodites at the usual time of reproduction by selfing and the viability of their progeny until the L1 larval stage. The log-transformed, covariate-adjusted self-fertility values (best linear unbiased prediction estimates, BLUPs) for each inbred line were downloaded from *Chelo et al., 2019*, exponentiated, and divided by the mean to obtain a proxy for relative fitness (noted $w$; *Table 1*).

## Phenotypic plasticity and phenotypic divergence

We used a multivariate analysis of variance (MANOVA) to model ancestral phenotypic plasticity and the divergence of locomotion behavior and body size. The six transition rates and body sizes were fitted as a multivariate response variable, with fixed effects of temperature and humidity at the time of movie recording and the log of object density in each Petri plate. These three environmental variables were centered and standardized before the analysis to mean=0 and sd=1. We further modeled a fixed effect of block and a fixed effect of year accounting for when the different lines were measured. The main factors of interest were the fixed effects of the salt environment and the fixed effects of evolution, together with their interaction; the last factor with four population ID levels (A6140, GA[1,2,4]50). The residual error was assumed to follow a multivariate normal distribution. We used the *manova*

function in the *stats* package in R for computation (***R Development Core Team, 2018***), with Wilks tests are being used for the significance.

From the MANOVA results, we extracted the Sums-of-Squares and Cross-Products (SSCP) matrices for the fixed effects of environment and population and eigendecomposed these matrices to describe the orthogonal canonical traits maximizing phenotypic variation in each (***Walter, 2023***). For the SSCP-matrix of the environment, the first eigenvector is the dimension containing the most phenotypic plasticity in the ancestral population and is here named $\delta p$. For the SSCP-matrix of evolution, the first eigenvector is the dimension of divergence among the four populations in high salt and is here called $d_{max}$ are the eigenvalues measuring the variation explained by each eigenvector. Estimated mean-least square divergence per replicate population is here called $\Delta \bar{q}_k$, with $k$ being the environment.

Additionally, we modeled the traits individually using linear mixed-effects models to estimate the best linear unbiased predictions (BLUPs) of transition rates and body size per inbred line (used only for visualization purposes in the figures). This univariate approach allowed testing the divergence of transition rates for each replicate GA population from the ancestral A6140 population. This univariate model was similarly formulated as the MANOVA, except the block was included as having random effects. For model fitting, we employed the *lme4* package in (***Bates et al., 2015***) in R. Post-hoc pairwise contrasts employed Tukey tests with the *emmeans* package (***Lenth, 2021***).

## G-matrices and genetic divergence

Using the same model, we estimated the $G$-matrices of the ancestral population A6140 and the three evolved replicate populations GA[1,2,4] separately for the traits measured in the low and high salt environments.

The six transition rates and body size were fitted as a multivariate response variable column-vector $y$ in the model:

$$\boldsymbol{y} \quad = \mu + \sum_{n=1}^{7} \alpha \times [T, H, D] + \gamma + \zeta + \eta + \epsilon \tag{2}$$

where $\mu$ are the intercepts and $\alpha$ are the environmental fixed effects of temperature (T), humidity (H), and log density (D). We denote $[T, H, D]$ to simplify notation of the product ($\times$) among the environmental variables (fitting all three variables as fixed effects, the three two ways interactions, and the three-way interaction for a total of seven fixed effects). $\gamma$ was defined as the fixed effect of year when the assays were conducted, $\zeta \sim \mathcal{N}(0, \sigma^2)$ and $\eta \sim \mathcal{N}(0, \sigma^2)$ the random effects of line and block identity, respectively. $\epsilon \sim \mathcal{N}(0, \sigma^2)$ defines the residual error.

The $G$-matrix is half the line identity (co)variance matrix ($\zeta$), as we have measured homozygous diploid inbred lines and assume codominance. As estimated here, the broad-sense $G$-matrix should be an adequate surrogate for the narrow-sense $G$-matrix. This is because there is no inbreeding depression for self-fertility in high salt due to the self-fertilization of hermaphrodites from the experimental outbred populations (***Chelo et al., 2019***), and because, at least in low salt, we failed to detect average (genome-wide) directional dominance or epistasis when comparing the means of transition rates in the outbred populations with those among the inbred lines (***Mallard et al., 2022***).

Models were fit with the R package *MCMCglmm* (***Hadfield, 2010***). We used improper flat priors (nu=0). Model convergence was verified by visual inspection of the posterior distributions and an autocorrelation below 0.05. 100,000 burn-in iterations were done with a thinning interval of 2000 and over 2 million MCMC iterations. The A6140 $G$-matrix in high salt was estimated using different prior distributions, chosen among the literature as the most representative including parameter-expanded priors (see ***Figure 2—figure supplement 1***).

Because the variance estimates resulting from the *MCMCglmm* models are positive definite, null expectations for the $G$-matrices were obtained by randomizing 1000 times the phenotypic data set. Randomization was done by shuffling inbred line and block identities and refitting the model at each iteration (***Equation 2***). We then computed the posterior mode for each of the 1000 models to construct a null distribution of genetic variances.

Eigendecomposition of each $G$-matrix was done in R as above for phenotypic (co)variances. We define the main canonical dimension of genetic variation $g_{max}$ as the first eigenvector of the A6140 $G$-matrix (with $\lambda_{g_{max}}$ its eigenvalue). We calculate the angle between the two $g_{max}$ in high and low salt as the mean of the estimated posterior distribution modes (this angle is defined in the next section, see ***Equation 3***).

We used the random skewers method described by *Aguirre et al., 2014*; *Hine et al., 2009* to describe the genetic divergence during experimental evolution. In this method, random vectors are projected through the four *G*-matrices to estimate the genetic variance in all phenotypic directions (*Equation 5*, see below). Using the *G*-matrix posterior distributions, we tested for significant differences in genetic variance between matrices for each random vector. The vectors that showed a significant difference (i.e. no overlap between the 95% CI of the two matrices projected variance) were retained to construct an *R*-matrix with the (co)variances of differentiation. The eigendecomposition of the *R*-matrix then describes the canonical traits of genetic differentiation among the *G*-matrices. The first eigenvector the *R*-matrix is here called the vector of genetic divergence ($e_{max}$, *Table 1*) because the A6140 ancestral population drives most differentiation. An alternative *G*-matrix differentiation analysis can be done with the eigentensor approach (*Aguirre et al., 2014*). Eigentensor analysis of the four A6140, GA[1,2,4]50 *G*-matrices in high salt gave similar results (see methods and results in the GitHub appendix).

## Selection differentials

For the ancestral population A6140, we also computed the $G_{qw}$-matrices as defined in *Stinchcombe et al., 2014*, which is the *G*-matrix of the 6 traits of locomotion behavior and body size expanded to include self-fertility. The last column-vector entries of the $G_{qw}$-matrix are thus the covariances between traits and relative fitness, the selection differentials ($s_k$).

Different individuals in separate assays were measured for self-fertility and transition rates/body size. To assess for a statistical bias on selection differential estimates when using self-fertility BLUP estimates (*Hadfield et al., 2010*), we generated 500 ancestral $G_{qw}$-matrices in high salt with within-line self-fertility variability across the replicated measurements of transition rates. For each transition rate measurement (one per Petri dish, see above), one inbred line self-fertility value was sampled from a normal distribution using the line's mean and, as standard deviation, the standard error of the mean multiplied by $\sqrt{2}$. In each line mean self-fertility was calculated from multiple individuals (at least 22 and up to 85) and on average there are two transition rate values per line. Our protocol thus mimics a random split of self-fertility into two groups of identical size. The $G_{qw}$-matrix is stable to within-line self-fertility variation and subsequent analysis was done with the initial $G_{qw}$-matrix estimates. We also ensured that the *G*-matrix contained in the $G_{qw}$-matrix is similar to the one computed above for the ancestral population.

## Phenotypic and genetic alignments

We used the metrics introduced by *Noble et al., 2019* to compare the alignment of ancestral standing genetic variation with the first canonical dimension of phenotypic plasticity ($\delta p$), or with the first canonical dimensions of adaptive phenotypic ($d_{max}$) or genetic ($e_{max}$) divergence. The first metric is the angle between two vectors. The angle between the i-th eigenvector of the A6140 *G*-matrix, $g_i$, and $\delta p$ is defined as:

$$\Theta = \frac{180}{\pi} cos^{-1}\left(\frac{\delta p \cdot g_i}{\|\delta p\| * \|g_i\|}\right). \tag{3}$$

As both $g_i$ and $-g_i$ are eigenvectors of the *G*-matrix, $\Theta$ values between 90° and 180° were transformed so that $\Theta$ always remains between 0° and 90° ($\Theta'$=180°-$\Theta$, results from using $-g_i$ instead of $g_i$ in *Equation 3*). Angles comparing the alignment of the ancestral $g_i$ with the axis of phenotypic and genetic divergence were calculated, by replacing $\delta p$ in *Equation 3* with $d_{max}$ and $e_{max}$, respectively.

For each angle, we sampled the posterior distribution of the A6140 *G*-matrix to create a credible interval. $\delta p$ and $d_{max}$ were obtained as the first eigenvectors of the SSCP matrices from the MANOVA model, as described above. The null expectation for $\Theta$ is calculated as the angle between 1000 pairs of random vectors sampled from a uniform distribution $\mathcal{U}^7(-1, 1)$.

The second metric computes the proportion of ancestral genetic variance along the main canonical trait of ancestral phenotypic plasticity:

$$r = \frac{\delta p^T \cdot G \cdot \delta p}{\|\delta p\|^2} \tag{4}$$

where $p_{max}$ is replaced by $d_{max}$ when computing the proportion of ancestral genetic variance in the main canonical trait of phenotypic divergence in high salt, or by $e_{max}$ when relative to the main canonical trait of genetic divergence in high salt.

$\Pi$ is the ratio between the amount of genetic variance in $r$ that maximizes plasticity, phenotypic or genetic divergence, over the maximum possible amount of genetic variance in any phenotypic dimension ($\lambda_{g_{max}}$, the first eigenvalue of the $G$-matrix):

$$\Pi = \frac{r}{\lambda_{g_{max}}} \tag{5}$$

$\Pi$ values are comprised between 0 (no genetic variance along the plasticity/divergence canonical traits) and 1 (when the plasticity/divergence canonical traits contain all the genetic variance in $g_{max}$). The null distributions for $\Pi$ were obtained by randomizing 1,000 ancestral $G$-matrices through shuffling inbred line and assay block identities.

## Selection differentials and gradients

Selection differentials were estimated above with the ancestral matrix $G_{qw}$ in high or low salt as the genetic covariance between transition rates and body size with self-fertility in high salt ($s_k$). Comparing observed and expected responses to selection was done by estimating directional selection gradients using Lande's retrospective equation, equation 9 in *Lande, 1979*. This is unlike *Stinchcombe et al., 2014* or *Hajduk et al., 2020*, where phenotypic selection gradients were obtained by regression of fitness onto the traits, following *Lande and Arnold, 1983*. In our case, genetic selection gradients on each transition rate were defined as:

$$\beta_g = G^{-1}s_k \tag{6}$$

and the phenotypic selection gradients as:

$$\beta = G^{-1}\Delta q_k \tag{7}$$

The $G$-matrix of the ancestral population was assumed constant during experimental evolution. The credible intervals of both selection gradients were estimated by sampling the posterior distribution of the $G$-matrix, assuming fixed high salt $s_k$ in **Equation 6** or fixed $\Delta \bar{q}_k$ for each replicate population in **equation 7**. We have also obtained credible intervals for $\beta_g$ by sampling the $G$-matrix and the posterior distribution of $s_k$.

Whether selection theory predicts the direction of phenotypic evolution amounts to an alignment between expected and observed phenotypic divergence. We thus calculated the angle (as above, $\Theta$) between the selection differentials on transition rates and body size in high or low salt ($s_k$), with the observed phenotypic divergence in high or low salt ($\Delta \bar{q}_k$). The null expectations for the angle were obtained by calculating the angles between 1000 pairs of random vectors sampled from a uniform distribution $\mathcal{U}^7(-1, 1)$. Similarly, whether selection theory predicts the magnitude of phenotypic evolution in high salt can be calculated as the ratio of observed phenotypic divergence over selection differentials ($\Delta \bar{q}_k/s_k$). We sampled the posterior distribution of $G$-matrix for these comparisons to obtain credible intervals.

## Contrasts between posterior distributions

The 'significance' of the posterior mode estimates are based on its overlap with the posterior null distribution of the posterior modes (*Walter et al., 2018*). For all comparisons of posterior distributions, significance can be inferred when their 83% credible intervals do not overlap (*Austin and Hux, 2002*), assuming homoscedasticity.

## Archiving

Self-fertility data has been previously published by *Chelo et al., 2019*, and locomotion behavior data in low salt for the ancestral population in *Mallard et al., 2022*. New data (adaptation, locomotion behavior and body size in high salt), R code, and modeling results are in our GitHub repository and will be archived in a public repository upon publication.

## Acknowledgements

We thank I Chelo, H Gendrot, T Guzella, and L Noble for help with worm handling, data acquisition or data analysis; C Baer, C Dillmann, T Long, L Noble, J Pemberton, P Phillips, S Proulx, J Sztepanacz and P de Villemereuil for discussion; J Sztepanacz, G Walter, and reviewers for suggestions on improving the presentation of this study. Funding: This work was supported by the European Research Council (ERC-St-243285), the Agence Nationale pour la Recherche (ANR-14-ACHN-0032–01, ANR-17-CE02-0017-01), and the KITP Quantitative Biology program (National Science Foundation, PHY-1748958; Gordon and Betty Moore Foundation, 2919.02).

## Additional information

### Funding

| Funder | Grant reference number | Author |
| --- | --- | --- |
| European Research Council | ERC-St-243285 | Henrique Teotónio |
| Agence Nationale pour la Recherche | ANR-14-ACHN-0032-01 | Henrique Teotónio |
| Agence Nationale pour la Recherche | ANR-17-CE02-0017-01 | Henrique Teotónio |
| National Science Foundation | PHY-1748958 | Henrique Teotónio |
| Gordon and Betty Moore Foundation | 2919.02 | Henrique Teotónio |

The funders had no role in study design, data collection and interpretation, or the decision to submit the work for publication.

### Author contributions

François Mallard, Conceptualization, Data curation, Software, Formal analysis, Validation, Investigation, Writing - review and editing; Bruno Afonso, Data curation, Investigation; Henrique Teotónio, Conceptualization, Resources, Supervision, Funding acquisition, Investigation, Writing - original draft, Project administration

### Author ORCIDs

François Mallard ⬤ http://orcid.org/0000-0003-2087-1914
Henrique Teotónio ⬤ http://orcid.org/0000-0003-1057-6882

### Decision letter and Author response

Decision letter https://doi.org/10.7554/eLife.80993.sa1
Author response https://doi.org/10.7554/eLife.80993.sa2

## Additional files

### Supplementary files

• MDAR checklist

### Data availability

New data, R code for analysis and modeling results is freely accessible and can be found at https://github.com/ExpEvolWormLab/Mallard_Robertson (copy archived at *Mallard and Teotonio, 2023*).

The following previously published dataset was used:

| Author(s) | Year | Dataset title | Dataset URL | Database and Identifier |
|---|---|---|---|---|
| Chelo IM, Afonso B, Carvalho S, Theologidis I, Goy C, Pino-Querido A, Proulx SR, Teotónio H | 2019 | Genotype and phenotype data sets | https://figshare.com/articles/dataset/Data/8665661 | figshare, 10.6084/m9.figshare.8665661 |

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

# Appendix 1

## Appendix 1—key resources table

| Reagent type (species) or resource | Designation | Source or reference | Identifiers | Additional information |
|---|---|---|---|---|
| Strain, strain background (*C. elegans, male and hermaphro dite*) | A6140 | DOI: 10.1186/s12915-014-0093-1 | A6140 | Ancestor, outbred population |
| Strain, strain background (*C. elegans, hermaphro dite*) | A6140L# | DOI: 10.1534/ genetics.117.300406 | A6140L# | A6140 inbred lines |
| Strain, strain background (*C. elegans, male and hermaphro dite*) | GA150 | DOI: 10.1186/ s12915-014-0093-1 | GA150 | Outbred population |
| Strain, strain background (*C. elegans, hermaphro dite*) | GA150L# | DOI: 10.1534/genetics.117.300406 | GA150L# | GA150 inbred lines |
| Strain, strain background (*C. elegans, male and hermaphro dite*) | GA250 | DOI: 10.1186/s12915-014-0093-1 | GA250 | Derived from A6140 |
| Strain, strain background (*C. elegans, hermaphro dite*) | GA250L# | DOI: 10.1534/genetics.117.300 406 | GA250L# | GA250 inbred lines |
| Strain, strain background (*C. elegans, male and hermaphro dite*) | GA450 | DOI: 10.1186/ s12915-014-0093-1 | GA450 | Outbred population |
| Strain, strain background (*C. elegans, hermaphro dite*) | GA450L# | DOI: 10.1534/genetics.117.300406 | GA450L# | GA250 inbred lines |
| Strain, strain background (*C. elegans, hermaphro dite*) | EEV1402 | DOI : 10.1038/ncomms3417 | EEV wormbase lab line 1402 | A6140 inbred line with GFP transgene ccIs4251 |
| Software, algorithm | MTW | DOI: 10.1038/nmeth.1625 | - | - |
| Software, algorithm | R | http://www.Rproject.org | - | version 3.3.2 |
| Software, algorithm | RStan | http://mc-stan.org/ | - | R package version 2.18.2 |
| Software, algorithm | stats | https://www.Rproject.org/ | - | R package version 3.3.2 |
| Software, algorithm | lme4 | doi: 10.18637/jss.v067.i01 | - | R package version 1.1-32 |
| Software, algorithm | emmeans | doi:10.1080/00031305.1980.10483031 | - | R package version 1.7.1-1 |
| Software, algorithm | multcomp | 10.1002/bimj.200810425 | - | R package version 1.4-23 |
| Software, algorithm | msm | DOI: 10.18637/jss.v038.i08. | - | R package version 1.7 |
| Software, algorithm | MCMCgl mm | DOI: 10.18637/jss.v033.i02. | - | R package version 2.34 |
| Software, algorithm | R scripts | https://github.com/ExpEvolWormLab/ Mallard_Robertson | - | This paper |

