## [Editor Report]

This is an important paper that takes advantage of a comprehensive evolutionary genetic dataset to tease apart the relationship between genetic variation, selection, and phenotypic divergence over 50 generations. The evidence supporting the conclusions is robust and aligns with a growing body of work that shows patterns of variation can predict divergence over long periods of time and also that evolution does not always occur in the direction of selection, particularly when selection is acting on genetically correlated traits. The questions addressed in this study will particularly appeal to evolutionary biologists and quantitative geneticists.

---

## [Decision Letter]

**Decision letter after peer review:**

Thank you for submitting your article "Selection and the direction of phenotypic evolution" for consideration by *eLife*. Your article has been reviewed by 3 peer reviewers, one of whom is a member of our Board of Reviewing Editors, and the evaluation has been overseen by Molly Przeworski as the Senior Editor. The following individual involved in the review of your submission has agreed to reveal their identity: Greg Walter (Reviewer #3).

Essential revisions:

1. Include body size as a trait in the analyses (these data seem to have already been collected, but not included).

2. Describe and justify what the movement traits represent biologically and why they would be adaptive in a high-salt environment.

3. Justify/explain why the movement traits can be considered independent traits and are not simply multiple measurements of the same phenotype.

4. Provide evidence that the fitness of these populations increased throughout this experiment.

5. Provide evidence and clarification of the method used to determine statistical support for genetic variance in G (showing that G has more than 1 significant genetic dimension would help provide support for #3).

6. Justify why heritability is an appropriate scale of measurement, rather than raw genetic variances or mean-scaled genetic variance.

7. Revise the introduction to provide targeted background on the key questions addressed.

8. Re-write the methods and results to improve clarity.

*Reviewer #1 (Recommendations for the authors):*

Line 63: I think you mean phenotypically correlated, not genetically correlated.

Line 68: the equation shown does not explicitly consider non-linear selection.

Line 90: under the infinitesimal model G is not predicted to change and the framework should generally work well across generations.

Line 141: the described methods do not match what is said in the rest of the paper. These methods say that the evolved populations were evolved in 8mM of salt for 35 generations and then kept at 305mM of salt for 15 generations. The rest of the paper implies that the concentration of salt gradually increases throughout the selection experiment. Which is it?

Line 160: line-mean fitness values are estimated with error. Are these errors incorporated into the models?

Line 235: how informative was the prior that was used, and how sensitive are the estimates to changing the prior? It seems that there are not enough genetic degrees of freedom to estimate all of the parameters in the model from the data.

Line 245: how did you estimate the null distributions? This is not explained.

Line 255: it's not obvious that the null expectation is 45. Could you show this empirically using bootstrapping to generate the null distribution.

Line 257: 1/6 of the total genetic variance is not the appropriate "null" distribution. Due to sampling error alone, the genetic variance will decline exponentially across the eigenvectors of G (see McGuigan and Blows 2015; Sztepanacz and Blows 2017).

Line 295: This does not seem like the correct comparison because of the potential for founder effects when establishing GA[1,2,4].

Tables and Figures: it looks like heritability and not genetic variances are reported. This is an important distinction

*Reviewer #2 (Recommendations for the authors):*

In addition to the previous comments of support for the overall question addressed, and the breadth of data available, but concerns over the evidence of adaptive evolution and multivariate nature of the traits, I had several further requests for clarification or comments about the interpretation of evolutionary genetic concepts.

Overall, I found the logical arguments presented in the introduction hard to follow, and what the study aimed to address was a surprise to me when I arrived at the end.

– Consider a general restructuring of the information to make it clearer from the start what the major focus of the study is.

– A lot of ground is covered very quickly, leaving the reader to fill in some big gaps from their own knowledge – consider whether all topics are sufficiently important to introduce up front, or if you can focus on the key ones to set the stage.

– Adding to the challenge of following the logic of the study motivation is that each individual sentence presents multiple, interconnected ideas. Some simple editorial changes that limited each sentence to one or two ideas I think would really help.

More specifically:

– Line 45: why does plasticity need to align with axes of genetic variance for adaptation to occur? The orientation of G and selection still matters – which you seem to also swing back to in the final sentence. It's not clear to me what this entire paragraph adds to the argument already made other than that phenotypic plasticity may alter the selection a population experiences when the environment changes.

– Line 53: I disagree that an adaptive argument is the most plausible or parsimonious here. The simpler explanation is that the G and environment are channeled through the same developmental pathways, and thus generate similar variation (i.e., Cheverud's argument for P=G due to alignment of G and E).

– Line 58: I don't understand how you arrive at the conclusion that when plasticity isn't adaptive it means we've misunderstood selection. Why is a presumption that plasticity must be adaptive warranted?

– Line 62: This is a misunderstanding of selection and evolution. Genetic correlations have no relevance for estimating selection, which acts solely on the phenotypic variation, irrespective of the causes of that variation. Genetic correlations will impact response to selection (i.e., evolution), but not selection itself.

– Why does it matter if evolution is due to selection acting directly on a trait or due to the genetic correlation of that trait to fitness (or another trait contributing to phenotypic differences in fitness)? How does whether the selection is direct or indirect have any bearing on whether plastic responses are adaptive? The genetic variance in plastic responses? Whether G is aligned with the direction of selection? How G evolves?

Some further requests for clarification of information:

– Equation 1 (and subsequent models): what is "CG"?

– Why are plasticity and divergence modeled for each trait individually? G is estimated from a multivariate model, so why are plasticity and divergence compiled as a vector of individual estimates?

– Equation 2 and 3: what is "Div".

– Equation 6: why is there a single intercept for the 6 traits? Why would traits not have their own individual intercept? Centering the data prior to analysis will not preclude traits differing in intercept when the fixed effects are taken into account.

– Line 243: why is e11 defined as the dimension with the most divergence?

– Line 248: "strictly" not "strickly".

– Line 251: Please explain what Gqw is – the 13 trait G?

– Equation 8: what is this "G" / where does it come from? How much of a bias is introduced here by taking the inverse of a G where many dimensions have very low variance (only positive due to imposed constraints)?

– Line 256: given the size of your experiment, what is the expected variation (error) around 45 degrees for two unrelated vectors?

– Second line after Equation 10 – add "genetic" for clarity: "…maximum amount of genetic variance …".

– Table 2 please also provide the error estimates.

– Figure 2: Could you please provide further information on the approach employed by the cited Morrissey and Bonnet 2019 for establishing null expectations? I might have missed this in the methods, but the evidence that you actually have genetic variance is key to all conclusions in the paper, so being very clear about this evidence is important.

– Bottom panel in B -I presume the vertical line indicates 0, but the use of red here and for high salt is confusing.

– Line 281: It's not clear what you mean by "modular" – the definition that I am familiar with relates to the strength of correlation among one set of traits relative to their correlation with other traits, but does not depend on a shared direction of correlation (i.e., a module contains both positively and negatively associated traits, so long as those correlations are relatively strong). The observation that one (maybe 2) axes capture all variation suggests a single module (i.e. a single behavioral syndrome has been measured).

– Line 284: what is "rounder" and how do you determine it's not "important"?

– Figure 3: Where does the null expectation come from? Please report the actual numbers for observed and null (including CI – in text or table) as it is not possible to see the overlap on this small plot on the plot. The conclusion that phenotypic plasticity is aligned with one G and not the other is very strong given that it does not seem that you can actually tell any difference between the two estimates (beyond sampling error alone).

– Tables 3 and 4. I have no idea what is being shown here – please provide sufficient information to relate this back to the Methods and the models that were fit, and what null hypothesis the reported Χ2 is associated with.

– Line 328: what's the logic for concluding that genetic variance in fitness indicates a stressful environment? Perhaps such conclusions are better placed in the Discussion where they might be justified via further information.

– Line 368: where does the information on gene expression come from? How is gene expression restricted to active/still? Do you mean that there is only divergence in expression between these states? How does this explain why leaving the stationary state has a different sign of divergence from entering the stationary state or changing direction? These observations seem consistent with my earlier interpretation that there is a single behavioral trait being assessed here, not six.

– Line 377: how statistically robust are the estimates of mutational variance? Where there is no variance, zero covariance will be implicated. The strength of this evidence also speaks to the question of whether there is a single "trait" being assessed here.

– Line 383: will only facilitate adaptation if they are aligned with future directions of selection.

– Line 450: why do you expect that changes in locomotor behavior will be under direct selection here? If the environment was heterogeneous for salinity (or food), then there may be fitness benefits, but how does moving any particular way allow the worms to increase their survival or fecundity in a high-salinity environment? Again, how do these results reflect a response to selection versus neutral evolution?

– Line 480: more details on where these data on size, and these estimates are coming from would be appreciated.

*Reviewer #3 (Recommendations for the authors):*

While I think these data are very valuable and their results are likely robust, I found that some parts of the manuscript were difficult to follow. In particular, it would help if the methods and results could be described a bit more clearly. In my major points below I highlight the areas I struggled to follow, and provide some suggestions for how to better focus the conceptual framework.

I think the authors need to better refine the conceptual framework and clarify the hypotheses in the final paragraph of the introduction. My confusion is because I'm not sure if they are focussing on adaptive phenotypic responses to the novel salt environment (i.e. plastic and evolved movement towards an optimum phenotype), or to predict the direction of phenotypic evolution. If the former, then I think testing whether plasticity in the novel environment is non-adaptive or adaptive is important, and testing whether phenotypic evolution has occurred in the direction of gmax or phenotypic selection (i.e. the phenotypic optimum) is also important. However the focus seems to be the latter, which means that there are currently no hypotheses justifying the test of the amount of genetic variance in the direction of plasticity, and furthermore, direct vs indirect selection is not explicitly compared. Below I provide some suggestions on how the conceptual framework could be clarified, I hope these comments help.

1) The framing in the introduction could be improved. In particular, the first paragraph jumps from selection on multiple traits to mutation-selection balance and alignment with the selection surface. I found some of the sentences quite long and it took several reads to understand their meaning. Furthermore, I found that plasticity is not well-grounded in the conceptual framework throughout the introduction and is not closely linked to the focus of the manuscript (predicting phenotypic evolution). L.46 assumes that plasticity is adaptive in a novel environment, which is often not the case. If the authors want to test the role of plastic responses in persisting and then adapting to the novel environment, I think they need some way of quantifying adaptive vs non-adaptive plasticity as well as testing whether adaptation occurs in the direction of plasticity (I found some information in supplementary material, but the link with the main idea of predicting phenotypic evolution is not clear). If it is possible to estimate a phenotypic selection gradient, they could test whether plasticity is adaptive by how well it aligns with phenotypic selection. However, this is a slightly different topic to predicting phenotypic evolution, which I think is the main focus of the manuscript. Furthermore, the second paragraph ends with 'indicating that selection is often misunderstood', which is a little vague and does not connect plasticity to phenotypic evolution.

2) Adaptation is important for this study, but evidence that the evolved populations adapted to the high salt environment is not presented. On lines 114 and 411 there is a reference to another study, but I think the manuscript would benefit from a more detailed explanation (or some discussion) about adaptation to the stressful environment, especially if the proxy for fitness (fertility) did not show evidence of adaptation (L.411).

3) There is missing information in the methods.

a. What are the traits (transition rates)? What do they represent and how are they important for the salt environment? There is some information L.180-193, but there is no biological explanation of what was measured or why these traits were chosen. It is also stated elsewhere that these traits are independent, could the authors please clarify how they are independent given they seem to use some of the same information in their calculation? I have trouble understanding how (for example) SF is different from FS in Figure 1.

b. Is the data collected from the same experiment that is a reciprocal transplant of all populations in all treatments? The section describing fitness (L.155) makes it sound like they were from different experiments. Furthermore, the use of BLUPs as estimates of fitness (from another experiment) is worrying, as explained by Hadfield et al. (2010; https://doi.org/10.1086/648604). If this is the case, it would be better to estimate the additive genetic covariance between traits and fitness (rather than the BLUPs). This is stated later in the manuscript, but I'm confused about where estimates of fitness came from and how they were used. I apologise if I've misunderstood the methods.

c. L.246 the authors describe a null distribution, but how was this constructed? Morrissey et al. (2019; https://doi.org/10.1111/evo.13842) made an important suggestion for more conservative estimates of null G (also described in Hangartner et al. 2020; https://doi.org/10.1111/evo.13891).

4) The description of the statistical analyses is sometimes difficult to follow. There are (by necessity) many parameters estimated and some more justification throughout the methods would help.

a. Some clarity would help in the description of equations 1-3 as it is difficult to understand what the motivation is for each equation, or what they represent. Also, equations 4-5 could be easier to understand by including (in text) the definition of a plasticity vector from Noble et al. (2019): δ X = Xnn – Xnov, where X is the mean of each trait in the nonnovel and novel environments.

b. L.257 and Equation10-12: I think equation 10 represents the amount of genetic variance in the direction of plasticity, but as a proportion of total plasticity (i.e. the length of the plasticity vector), is this correct? If so, please clarify in the text. A minor point on L. 257: this equation represents the amount of genetic variance in the direction of plasticity, which is different to genetic variance in plasticity. I am also confused by equations 10-11, why not just calculate the proportion of genetic variance in the direction of maximum evolvability (i.e. denominator in equation 10 is the 1st eigenvalue of G) – with plasticity vector normalised to unit length so that it represents the direction of plasticity. This removes the need for equation 11 and gives you the same information: the amount of genetics in the direction of plasticity as a proportion of the direction of maximum genetic variance. I apologise again if I've misunderstood, but I think a more detailed description and justification for equations 10-11 would help.

c. Equation 12: I don't think this is the correct null expectation. Why would we expect plasticity to be in the direction of 'average' genetic variance? I would think a better null would be random vectors (of unit length) projected through G so that you would test whether plasticity is in a direction that describes greater genetic variance than expected by random sampling.

d. Equation 5: Is this calculated for each of the derived populations? This is important for understanding the results in Figure 5. Because they are independent populations (with G also estimated separately), to me it would make more sense to do pairwise comparisons for each derived population with the ancestral (which could be summarised in Figure 5).

e. Figure 6 and 7 (+ their interpretation): To test how to predict phenotypic evolution, I think it would be better to directly compare the distribution of β and the selection differentials. By comparing them separately it is not clear what hypothesis is being tested and the argument is verbal rather than quantitative – see Hajduk et al. (2020; https://doi.org/10.1098/rstb.2019.0359) for a nice example.

5) In the discussion, I found it very interesting that the authors found body size to be both genetically correlated to the movement traits, and that selection was predicted accurately (both sg and β). I am curious as to why this trait wasn't included in the analyses because, as the authors highlight, it is probably the trait selection is operating on (or at least provides an estimate of performance as outlined by the traditional Lande and Arnold selection analyses). It made me wonder how body size changed across treatments and whether it evolved in the high salt treatment. I would think that it would be important to include body size in the analysis.

6) The discussion could be a bit more concise. I also think alternative explanations need to be discussed as I'm not sure I agree with the interpretation on L.425. It is more parsimonious (or at least a viable alternative) that during adaptation genetic variance has been depleted as would be expected if the selection is strong. Another alternative would be that selfing in a stressful environment has reduced the amount of genetic variance. I think it could be worth including these alternative explanations.

[Editors' note: further revisions were suggested prior to acceptance, as described below.]

Thank you for resubmitting your work entitled "Selection and the direction of phenotypic evolution" for further consideration by *eLife*. Your revised article has been evaluated by Molly Przeworski (Senior Editor) and a Reviewing Editor.

Two reviewers and I have reviewed the substantial changes made to the manuscript since its previous submission. We are all in agreement that the current version adequately addressed previous concerns and is largely improved. However, the manuscript would benefit from some additional minor revisions in writing to help improve the clarity of this technically and biologically complex paper. I do not expect the changes should take the authors very long to complete.

There are several suggestions from the reviewers for ways to improve clarity. Addressing the premise of the study in a clearer and more biologically motivated way, and clarifying the methods and results for testing indirect versus direct selection are particularly important.

*Reviewer #2 (Recommendations for the authors):*

The responses from the Authors are comprehensive, and I believe have adequately addressed concerns about analyses and interpretation, as well as improving clarity and accessibility of the paper. In particular, the Authors have clarified the approach taken to defining a null distribution against which observed parameter estimates are compared. The shuffling of data and estimation of G from multiple shuffled samples characterized random associations in the data. The Authors have also included further empirical data that provides evidence that the three experimental populations had adapted to high salt.

Throughout the manuscript there is a clear commitment to transparency of data and analyses, with access to data and additional results to support the main results reported.

*Reviewer #3 (Recommendations for the authors):*

The authors present an interesting test of whether experimental evolution can be predicted from the patterns of genetic variation in the ancestral population. The authors have done an impressive job to address the comments raised in the previous review, which means that the paper is much stronger and focused on the questions addressed. I appreciated their detailed responses to the questions raised in the previous review and I found the new version greatly improved.

I only have one comment: The writing, while comprehensive, is quite dense and often abstract throughout the paper. This will make it difficult for a broader audience to follow. In addition, because of the (necessary) statistical rigor, it felt like the biology is missing, especially in the setup of the study and the results. For example, terms such as 'canonical traits' and references to alignments in the first paragraph make it difficult for a more broader audience to understand the background of the study, or the gap in knowledge that is being addressed.

I provide some specific examples below, but suggest that revising should focus on clarity throughout (and on the biological rather than statistical importance):

– L29-33 it seems early in the introduction to introduce G and Lande's equation, and it does so with little biological foundation.

– L.13 a better way of saying 'follow their selection gradient' would be 'whether phenotypic evolution occurs in the direction of selection'.

– L.14 'the canonical trait of the multivariate phenotype' this is not broadly accessible. Perhaps 'trait combinations with large amounts of genetic variation' (or something similar) would be more intuitive.

The premise of the study is much clearer but could be clarified further. The paragraph in the introduction L.115-116 needs to be simplified as it is difficult to follow and just lists the contrasts in a complicated way. Instead, it would be better to include the tests and hypotheses, rather than 'characterising' adaptation. The final two sentences are confusing (L.123-127), they should more clearly outline the focus of the study. In particular, I'm not sure what the final sentence is saying: are you testing whether genetic selection gradients in the ancestral population = phenotypic selection gradients after adaptation? I would have thought you would compare genetic selection gradients with phenotypic divergence between the ancestral and adapted populations.

I liked the addition of the explanation of the traits in the introduction, but is it possible to add some more biology? L.111-112 '… which ones [traits] are genetically or environmentally independent' is a bit vague. Do you instead have an idea of how the movement traits help them to navigate low vs high salt environments? Or some other more biological reasoning. It is suggested 'movement can increase during experimental evolution due to more foraging and dwelling' – but this is confusing, is it that there is selection for greater movement across generations? Or just that there is greater movement in a new environment? Foraging and dwelling seem like opposites unless dwelling is defined as a specific behaviour.

I found the major tests of indirect vs direct selection and also how phenotypic evolution is predicted very difficult to follow in the results. I think the analyses are correct, but the results need to be described more simply so that it is easier to understand. I'm not sure the section on direct selection is required, and if removed, could help simplify the study to focusing on predicting phenotypic evolution. Or there needs to be a better justification for including direct selection because the Stinchombe (2014) and Hajduk et al. (2020) framework does not seem to have been used (but I apologise if I've misinterpreted something). Specifically, is there a reference for equation 7? Why does this Β represent phenotypic selection when it seems to capture phenotypic divergence. I am also a little confused as to why selection differentials are not just compared to observed divergence. And if you want to include direct vs indirect, predicted evolutionary change with indirect selection could be calculated using δ Z = G*Β where Β is calculated as per equation 6 – this uses the Stinchcombe (2014) framework. Sorry again if I've misinterpreted something, but there seems to be a disconnect between the results as written, the figures and the interpretation seems contradictory in the abstract (L.15-16).

Overall I found the study very interesting, I hope my comments help.

---

## [Author Response]

Essential revisions:1. Include body size as a trait in the analyses (these data seem to have already been collected, but not included).

We have included body size in all the analyses as the seventh trait (the other six being the transition rates between movement states). We obtain our estimates separately for each of the two environments (the low salt domestication environment and the high salt novel environment). To estimate selection differentials in the ancestral population, we estimate a Gmatrix with 8 traits (G_qw_-matrix), including transition rates, body size and self-fertility. As suggested by the reviewers, we now include multivariate analyses of phenotypic variation (MANOVA). We keep, however, univariate phenotypic analysis as they allow us to estimate mean values for each inbred line (for visualization purposes) and also test the significance of phenotypic divergence for each of the replicate evolved populations. Multivariate and univariate approaches give similar results however.

2. Describe and justify what the movement traits represent biologically and why they would be adaptive in a high-salt environment.

We have one paragraph in the Introduction that describes and justifies why transition rates and body size might be relevant.

3. Justify/explain why the movement traits can be considered independent traits and are not simply multiple measurements of the same phenotype.

This is also justified in the Introduction (and Methods, equation 2). In the Results section we show that genetically, there are 3 canonical traits for locomotion behavior (eigenvectors that explain a significant amount of ancestral genetic variance). We now also provide a summary Table 3, which relates the main canonical traits of standing genetic variation with those of ancestral phenotypic plasticity, of phenotypic and genetic divergence and of mutational covariance that we recently published (Mallard et al. G3, 2023). Finally, in the Discussion, one paragraph justifies why more than one trait can be independent given standing variation, mutation and selection.

4. Provide evidence that the fitness of these populations increased throughout this experiment.

We did new experiments to show adaptation to the high salt environment. The results confirm adaptation after 50 generations of evolution, for all replicate populations (Figure 5).

5. Provide evidence and clarification of the method used to determine statistical support for genetic variance in G (showing that G has more than 1 significant genetic dimension would help provide support for #3).

Please see the Methods section: we repeatedly shuffled the phenotypic dataset by line and block identity to obtain a null posterior distribution of the genetic variance modes at each iteration. Whenever this approach was used we label it in orange (figures), so hopefully it will be clearer for the reader. We believe that our sampling procedure addresses the problem you worked on in Sztepanacz and Blows (Genetics, 2017). The first reason is that to obtain uncertainties we compared the G-matrix with the distribution of posterior modes of random Gmatrices. The second reason is that we are also asking the question of whether genetic variances in the observed eigentraits of the ancestral population are different from null expectations, and thus conditional on the how the traits are genetically structured. For this, we projected the random G-matrices onto the observed eigenvectors (Figure 2, supplement Figure 3). Both approaches return similar results for the expected null distribution. Please let us know if you think we are wrong so that we can correct it. Also note that we do sample the posterior of the empirical G-matrix when estimating the selection gradients (Figure 9).

6. Justify why heritability is an appropriate scale of measurement, rather than raw genetic variances or mean-scaled genetic variance.

There was a misunderstanding. We did not present heritability estimates nor we do it now. We only present estimates of genetic (co)variances throughout. As suggested by reviewer #1 we now assess different prior distributions when estimating the G-matrix (Figure 2, figure supplement 1). The inclusion of body size data forced us to also consider scaling issues: we have opted for multiplying body size data by 50 in order to be on a similar phenotypic scale as the transition rates in movement states.

7. Revise the introduction to provide targeted background on the key questions addressed.

We have completely revised the Introduction, hopefully it will read much better. We are nonetheless somewhat terse when considering short-term versus long-term evolutionary dynamics, though we would like to keep it that way because we think it is important to understand how theory assumptions, and existing comparative evidence, are limited.

8. Re-write the methods and results to improve clarity.

Done. We have also included several subsection headings to improve clarity. Please see reply to the reviewers where we detail new methods, analyses and results.

Reviewer #1 (Recommendations for the authors):9. Line 63: I think you mean phenotypically correlated, not genetically correlated.

Thank you. Corrected.

10. Line 68: the equation shown does not explicitly consider non-linear selection.

Thank you. Corrected.

11. Line 90: under the infinitesimal model G is not predicted to change and the framework should generally work well across generations.

Indeed. Corrected.

12. Line 141: the described methods do not match what is said in the rest of the paper. These methods say that the evolved populations were evolved in 8mM of salt for 35 generations and then kept at 305mM of salt for 15 generations. The rest of the paper implies that the concentration of salt gradually increases throughout the selection experiment. Which is it?

It is the first regime. We have throughout tried to make it explicit.

13. Line 160: line-mean fitness values are estimated with error. Are these errors incorporated into the models?

This is indeed a concern. The inbred self-fertility (fitness) are estimated with error but it was not incorporated in any model. We now provide an additional analysis to test whether there is a consistent bias in the genetic covariance estimates between traits and fitness. We generated for each line a fertility value randomly sampled from a normal distribution centered on the observed line mean and presuming that the initial set of individuals used to estimate fitness was randomly split in two groups (multiplying by two the expected variance) as we have two technical replicates per line for phenotyping. We show that the mean estimates of selection differentials are robust to within line fertility variation (Figure 4, supplement figure 1).

14. Line 235: how informative was the prior that was used, and how sensitive are the estimates to changing the prior? It seems that there are not enough genetic degrees of freedom to estimate all of the parameters in the model from the data.

We provide more details on the prior used to compute the G-matrices (flat improper prior). We further show the ancestral G-matrix estimates using different priors, including parameter expanded priors (Figure 2, supplement figure 1). Overall, ancestral G-matrix estimates are robust to varying prior distributions, particularly the covariance estimates. For the variance estimates we find that when using the Inverse-Wishart distribution as a prior with a high “degree of belief”, only body size has a lower value than when using other priors. It is unclear to us if this is much of an issue for subsequent analysis. We do, however, conclude that are enough degrees of freedom to model G-matrices (see also replies regarding how uncertainties are found).

15. Line 245: how did you estimate the null distributions? This is not explained.

The null distributions involving the randomization of line and block IDs, reestimation of the G-matrix, and eigen decomposition where appropriate, are shown with orange labels. In the methods we explain the approach in the Methods and then in the figure captions when necessary.

16. Line 255: it's not obvious that the null expectation is 45. Could you show this empirically using bootstrapping to generate the null distribution.

We had trouble understanding the null of this angle as well and really appreciate the reviewer's input as the null expectation is not 45°. It will only be 45° in two dimensions and progressively approaches 90° as the number of dimensions increases. We now generate a null distribution using bootstrapping as you suggest. We failed to find an appropriate reference for this problem, or the ones we found were mathematically challenging. Please see Methods.

17. Line 257: 1/6 of the total genetic variance is not the appropriate "null" distribution. Due to sampling error alone, the genetic variance will decline exponentially across the eigenvectors of G (see McGuigan and Blows 2015; Sztepanacz and Blows 2017).

We agree that the genetic variance will decline exponentially even with randomly generated trait values (though it will occur in random directions). We now provide a null which is based on the randomized G-matrices obtained by shuffling line and block IDs. In Figure 2, supplement figure 3 we also present a null conditional on the observed eigentraits of the ancestral population. The observed values are lower than both kinds of nulls which strengthen our conclusion that 3 traits are genetically orthogonal in the ancestral population regardless of the salt environment. Please see also reply to point #5.

18. Line 295: This does not seem like the correct comparison because of the potential for founder effects when establishing GA[1,2,4].

There were very limited founder effects as the GA replicates were split equally at sample sizes of at least the effective population size in low salt, from the same unique ancestor population (A6140). If anything there was reduced genetic variance in the ancestor, before deriving the replicates, as at the time of phenotyping it had spent more time under cryogenesis. We do not wish to detail potential problems with our protocols as their relevance for experimental evolution can be found in our previous work (see for example, Guzella et al. PlosGenetics 2018 for the consequences of drift and bottlenecks on adaptation to high salt).

19. Tables and Figures: it looks like heritability and not genetic variances are reported. This is an important distinction

No heritability estimates were presented, only genetic variances.

Reviewer #2 (Recommendations for the authors):I have several requests for clarification or comments about the interpretation of evolutionary genetic concepts.Overall, I found the logical arguments presented in the introduction hard to follow, and what the study aimed to address was a surprise to me when I arrived at the end.20. Consider a general restructuring of the information to make it clearer from the start what the major focus of the study is.21. A lot of ground is covered very quickly, leaving the reader to fill in some big gaps from their own knowledge – consider whether all topics are sufficiently important to introduce up front, or if you can focus on the key ones to set the stage.22. Adding to the challenge of following the logic of the study motivation is that each individual sentence presents multiple, interconnected ideas. Some simple editorial changes that limited each sentence to one or two ideas I think would really help.

We revised the whole manuscript, and particularly the text in the Introduction. We have removed the paragraph on phenotypic plasticity (and now only interpret the phenotypic divergence and adaptation in light of ancestral phenotypic plasticity in the Discussion). The focus of the study is on the predictability of multivariate trait evolution, and thus on Lande’s equation when using selection differentials to estimate the directional selection gradients.

More specifically:23. Line 45: why does plasticity need to align with axes of genetic variance for adaptation to occur? The orientation of G and selection still matters – which you seem to also swing back to in the final sentence. It's not clear to me what this entire paragraph adds to the argument already made other than that phenotypic plasticity may alter the selection a population experiences when the environment changes.

We’ve eliminated the introductory paragraph on phenotypic plasticity.

24. Line 53: I disagree that an adaptive argument is the most plausible or parsimonious here. The simpler explanation is that the G and environment are channeled through the same developmental pathways, and thus generate similar variation (i.e., Cheverud's argument for P=G due to alignment of G and E).

We agree, but no longer refer to Cheverud’s original idea as we do not wish to discuss standing environmental stochastic variation, nor its evolution. We have little power to do so, and, more importantly, our phenotyping design and modeling not allow us to separate the effects of random environmental covariantes (error) from those of within-genotype stochastic variance. In our populations, locomotion behavior appears to be phenotypically and genetically similarly aligned (not shown). This question also relates to how many genetically independent traits exist, in the sense that adaptive phenotypic evolution could also have been limited by g2 or g3 but again we lack power to test their influence. The result that the selection gradient does not appear to align with neither standing genetic variation or mutational variance (Table 3) further suggests that there are fundamental developmental or physiological constraints. We now discuss this in the context of phenotypic plasticity and the evolution of novelty during adaptation, though without explicitly referring to Cheverud’s idea.

25. Line 58: I don't understand how you arrive at the conclusion that when plasticity isn't adaptive it means we've misunderstood selection. Why is a presumption that plasticity must be adaptive warranted?

The sentence was eliminated. Please see Discussion. Ancestral phenotypic plasticity is “non-adaptive” and only reveals the topography of the adaptive landscape.

26. Line 62: This is a misunderstanding of selection and evolution. Genetic correlations have no relevance for estimating selection, which acts solely on the phenotypic variation, irrespective of the causes of that variation. Genetic correlations will impact response to selection (i.e., evolution), but not selection itself.

Rephrased.

27. Why does it matter if evolution is due to selection acting directly on a trait or due to the genetic correlation of that trait to fitness (or another trait contributing to phenotypic differences in fitness)? How does whether the selection is direct or indirect have any bearing on whether plastic responses are adaptive? The genetic variance in plastic responses? Whether G is aligned with the direction of selection? How G evolves?

Correct. We’ve eliminated introducing plasticity upfront. We consider that a trait is adaptive, regardless of the environment where it was measured, whenever mean trait changes are correlated with adaptation to the target environment (high salt). We do not estimate ancestral genetic variance for plasticity (as it is collinear with variance in one of the environments, assuming linear reaction norms) nor do we explicitly test for the evolution of plasticity. See above replies and Discussion.

Some further requests for clarification of information:28. Equation 1 (and subsequent models): what is "CG"?

It was referring to “Common Garden” with fixed levels correcting for differences between the years of phenotyping locomotion behavior. We simplified the presentation of the model (Equation 2).

29. Why are plasticity and divergence modeled for each trait individually? G is estimated from a multivariate model, so why are plasticity and divergence compiled as a vector of individual estimates?

We model plasticity and phenotypic divergence in a single model using a multivariate model. See reply to point #1.

30. Equation 2 and 3: what is "Div".

Deleted.

31. Equation 6: why is there a single intercept for the 6 traits? Why would traits not have their own individual intercept? Centering the data prior to analysis will not preclude traits differing in intercept when the fixed effects are taken into account.

The phenotypic data are not centered and each trait has its own intercept (µ is a column vector). We center and standardize the design and environmental covariates.

32. Line 243: why is e11 defined as the dimension with the most divergence?

We have replaced the eigentensor analysis with a random skewers approach, as it simplifies the interpretation of “genetic divergence”. Please see Methods, and Figure 7, panel C. Briefly, both the random skewers approach and the previous eigentensor approach (now presented only in our GitHub repository) compare the four G-matrices and describe differentiation among them. Eigen decomposition of the (co)variances matrices for differentiation reveals that there is only one canonical trait explaining most variation which is due to differences of the evolved replicate populations to the ancestor population. For this reason, we call this canonical trait the dimension of most divergence (and not differentiation).

33. Line 248: "strictly" not "strickly".

Corrected.

34. Line 251: Please explain what Gqw is – the 13 trait G?

Following reviewer #3 suggestion, we estimate G_qw_ using the 6 transition rates and body size measured in high salt or low salt, together with the high salt fertility values. There are therefore 8 traits for each Gqw.

35. Equation 8: what is this "G" / where does it come from? How much of a bias is introduced here by taking the inverse of a G where many dimensions have very low variance (only positive due to imposed constraints)?

Thanks for this comment. All the trait variances are significantly above the null expectations (the randomized G matrices; Figure 2, supplement figure 2), but clearly two traits have very low variance (FS, BS). To address if taking the inverse of these variances leads to a bias in the selection gradients estimates, we repeated the analysis but without including those two traits (red in Author response image 1). Furthermore, we also did the analysis per trait, ignoring all other traits (green in Author tesponse image 1). There is no apparent bias as the original estimates with all traits included (gray) are similar when including or not the two low variance traits. The univariate approach (ignoring covariances between traits) obviously results in similar estimates as the selection differentials (compare Author response image 1 with Figure 4 in main text). We agree that the selection gradient for FS and BS are likely to be biased as a large proportion of the observed genetic variance is also detected in the randomized G-matrices. Yet, in the multivariate case, the selection gradients are not different from zero and their low values seem to be associated with an increased CI. This analysis in Figure 9, supplement figure 1.

**Author response image 1. sa2fig1:** Selection gradients estimated with all 7 traits (gray, as shown in Figure 9 of the manuscript), a subset of 5 traits (red) or separately (green) using ββgg = GG^-1^*s_k_*. CI are all obtained from sampling the posterior of the ancestral G-matrix and using fixed selection differentials.

36. Line 256: given the size of your experiment, what is the expected variation (error) around 45 degrees for two unrelated vectors?

We replaced this null expectation as mentioned in the response to reviewer #1 (reply #23). We produced random expectations based on the sampling of pairs of vectors from a uniform distribution.

37. Second line after Equation 10 – add "genetic" for clarity: "…maximum amount of genetic variance …".

Thank you.

38. Table 2 please also provide the error estimates.

Table 2 has been replaced by several tables with the results of the MANOVA and univariate analysis. Errors for each replicate population response could only be obtained from univariate models (used just for plotting in Figure 6). The comparison of each replicate with the ancestral population is shown in the Figures 1 and 6 as well as in supplementary tables. For the MANOVA analysis, each trait is contrasted between replicate populations and the ancestral population (Figures 1 and 6, lines; and supplementary tables). The major factor results are presented in Table 2 (the full results as a supplementary table).

39. Figure 2: Could you please provide further information on the approach employed by the cited Morrissey and Bonnet 2019 for establishing null expectations? I might have missed this in the methods, but the evidence that you actually have genetic variance is key to all conclusions in the paper, so being very clear about this evidence is important.

See replies #5, #12, and #18.

40. Bottom panel in B -I presume the vertical line indicates 0, but the use of red here and for high salt is confusing.

The vertical line is in black in all plots now.

41. Line 281: It's not clear what you mean by "modular" – the definition that I am familiar with relates to the strength of correlation among one set of traits relative to their correlation with other traits, but does not depend on a shared direction of correlation (i.e., a module contains both positively and negatively associated traits, so long as those correlations are relatively strong). The observation that one (maybe 2) axes capture all variation suggests a single module (i.e. a single behavioral syndrome has been measured).

We agree that the use of the term modular was irrelevant. We eliminated this discussion altogether.

42. Line 284: what is "rounder" and how do you determine it's not "important"?

We have rephrased the sentence and eliminated these terms.

43. Figure 3: Where does the null expectation come from? Please report the actual numbers for observed and null (including CI – in text or table) as it is not possible to see the overlap on this small plot on the plot. The conclusion that phenotypic plasticity is aligned with one G and not the other is very strong given that it does not seem that you can actually tell any difference between the two estimates (beyond sampling error alone).

The canonical trait of ancestral phenotypic plasticity is derived from the MANOVA analysis (pmax). Projections of the G matrix on pmax, and the angle between gmax, g2 and g3 with pmax are presented only for the high salt environment (Figure 3). The null for the projections comes from randomizing the G matrix, for the angles from sampling pairs of random vectors. We have added this explanation and the CI values in the figure caption (and also as a supplementary table). Similarly, we have provided all relevant information in Figure 8.

44. Tables 3 and 4. I have no idea what is being shown here – please provide sufficient information to relate this back to the Methods and the models that were fit, and what null hypothesis the reported Χ2 is associated with.

Most of the phenotypic analyses have been modified. We hope that the results are more clearly presented. Please note that all data, tables and scripts for analysis can be found by following the links provided in the figure legends. The analysis and results should be fully reproducible by the reader.

45. Line 328: what's the logic for concluding that genetic variance in fitness indicates a stressful environment? Perhaps such conclusions are better placed in the Discussion where they might be justified via further information.

We no longer use the term “stressful”.

46. Line 368: where does the information on gene expression come from? How is gene expression restricted to active/still? Do you mean that there is only divergence in expression between these states? How does this explain why leaving the stationary state has a different sign of divergence from entering the stationary state or changing direction? These observations seem consistent with my earlier interpretation that there is a single behavioral trait being assessed here, not six.

We have eliminated most of this paragraph.

47. Line 377: how statistically robust are the estimates of mutational variance? Where there is no variance, zero covariance will be implicated. The strength of this evidence also speaks to the question of whether there is a single "trait" being assessed here.

We no longer explicitly refer to pleiotropy and instead only mention that the first two traits with most mutational variance are not the same as the one under directional selection, or as the one with most ancestral standing genetic variation. Please see Table 3. Our estimates of mutational variance are statistically robust (DOI: 10.1093/g3journal/jkac335).

48. Line 383: will only facilitate adaptation if they are aligned with future directions of selection.

Indeed, rephrased.

49. Line 450: why do you expect that changes in locomotor behavior will be under direct selection here? If the environment was heterogeneous for salinity (or food), then there may be fitness benefits, but how does moving any particular way allow the worms to increase their survival or fecundity in a high-salinity environment? Again, how do these results reflect a response to selection versus neutral evolution?

Mechanistically, we don’t know why locomotion behavior would change in high salt. We briefly introduce the rationale to study locomotion, but one of the goals of our study is precisely to find how many traits are environmentally or genetically independent. What we show is that changes in locomotion behavior are adaptive in high salt and result from a response to indirect or direct selection. Further investigation will be necessary to find the “biological” relevance of these mathematically defined traits, in particular by mapping QTLs for the several canonical traits we describe. See also replies #4 and #27, for the adaptive nature of phenotypic evolution.

50. Line 480: more details on where these data on size, and these estimates are coming from would be appreciated.

We now include data on body size in all analyses.

Reviewer #3 (Recommendations for the authors):51. While I think these data are very valuable and their results are likely robust, I found that some parts of the manuscript were difficult to follow. In particular, it would help if the methods and results could be described a bit more clearly. In my major points below I highlight the areas I struggled to follow, and provide some suggestions for how to better focus the conceptual framework.I think the authors need to better refine the conceptual framework and clarify the hypotheses in the final paragraph of the introduction. My confusion is because I'm not sure if they are focussing on adaptive phenotypic responses to the novel salt environment (i.e. plastic and evolved movement towards an optimum phenotype), or to predict the direction of phenotypic evolution. If the former, then I think testing whether plasticity in the novel environment is non-adaptive or adaptive is important, and testing whether phenotypic evolution has occurred in the direction of gmax or phenotypic selection (i.e. the phenotypic optimum) is also important. However the focus seems to be the latter, which means that there are currently no hypotheses justifying the test of the amount of genetic variance in the direction of plasticity, and furthermore, direct vs indirect selection is not explicitly compared. Below I provide some suggestions on how the conceptual framework could be clarified, I hope these comments help.

Thank you for the time and effort you took in reviewing our manuscript. We believe to have addressed all your concerns and have followed your suggestions. Please see the new Introduction for an explanation of our focus of study (predicting phenotypic evolution). We do spend 2/3 of the results text, however, describing the ancestral states and evolution.

52. The framing in the introduction could be improved. In particular, the first paragraph jumps from selection on multiple traits to mutation-selection balance and alignment with the selection surface. I found some of the sentences quite long and it took several reads to understand their meaning. Furthermore, I found that plasticity is not well-grounded in the conceptual framework throughout the introduction and is not closely linked to the focus of the manuscript (predicting phenotypic evolution). L.46 assumes that plasticity is adaptive in a novel environment, which is often not the case. If the authors want to test the role of plastic responses in persisting and then adapting to the novel environment, I think they need some way of quantifying adaptive vs non-adaptive plasticity as well as testing whether adaptation occurs in the direction of plasticity (I found some information in supplementary material, but the link with the main idea of predicting phenotypic evolution is not clear). If it is possible to estimate a phenotypic selection gradient, they could test whether plasticity is adaptive by how well it aligns with phenotypic selection. However, this is a slightly different topic to predicting phenotypic evolution, which I think is the main focus of the manuscript. Furthermore, the second paragraph ends with 'indicating that selection is often misunderstood', which is a little vague and does not connect plasticity to phenotypic evolution.

We have re-written the Introduction. We first introduce Lande’s equation to predict phenotypic evolution, then the problem of indirect selection to estimate selection gradients. This is followed by a (necessarily) brief paragraph on why Lande’s equation might not be appropriate to predict phenotypic evolution (changing selection gradients and unstable Gmatrix over several generations) and finally a summary of how we present the experiment, phenotyping, and analyses. We have eliminated the paragraph on phenotypic plasticity and only discuss it at the end of the manuscript.

53. Adaptation is important for this study, but evidence that the evolved populations adapted to the high salt environment is not presented. On lines 114 and 411 there is a reference to another study, but I think the manuscript would benefit from a more detailed explanation (or some discussion) about adaptation to the stressful environment, especially if the proxy for fitness (fertility) did not show evidence of adaptation (L.411).

We performed an additional experiment to address this question. All four populations (ancestral and derived) were pairwise competed with a tester GFP-marked strain in the high salt environment, for one generation. Adaptation is measured by how much the experimental wild-type allele changes in this competition. We find adaptation to high salt (Figure 5).

54. There is missing information in the methods.a. What are the traits (transition rates)? What do they represent and how are they important for the salt environment? There is some information L.180-193, but there is no biological explanation of what was measured or why these traits were chosen. It is also stated elsewhere that these traits are independent, could the authors please clarify how they are independent given they seem to use some of the same information in their calculation? I have trouble understanding how (for example) SF is different from FS in Figure 1.

We have expanded the methods section to better explain how the transition rates were estimated. We have also added a paragraph to the Introduction addressing the potential biological significance of transition rates in the two salt environments. The transition rates are the probabilities that an individual will change its current state of movement given its state at an earlier time step. Transition rates were independently estimated (as opposed to the self-transition rates which are constrained to be a linear combination of two other transition rates, equation 1). The environmental mean distribution of SF and FS appears similar and they both increase with salt concentration (Figure 1). Genetically, however, they are negatively correlated with each other (Figure 2) and respond to selection in opposite directions (Figure 6).

b. Is the data collected from the same experiment that is a reciprocal transplant of all populations in all treatments?

We have tried to clarify our design in the Methods. Inbred lines from all four populations were assayed at the same time after experimental evolution, using frozen samples. Each “block” corresponds to a different sample thaw and phenotyping. Other environmental covariates were modeled, and importantly grandmaternal and maternal environmental effects were the same in each block. Each block contained several lines, usually from several populations, which were phenotyped in both low salt and high salt. The raw data table and sample size should also make our design clearer (presented as supplementary tables).

55. The section describing fitness (L.155) makes it sound like they were from different experiments. Furthermore, the use of BLUPs as estimates of fitness (from another experiment) is worrying, as explained by Hadfield et al. (2010; https://doi.org/10.1086/648604). If this is the case, it would be better to estimate the additive genetic covariance between traits and fitness (rather than the BLUPs). This is stated later in the manuscript, but I'm confused about where estimates of fitness came from and how they were used. I apologise if I've misunderstood the methods.

Yes, the fertility estimates were obtained in different assays. This is why we have used BLUPs, in order to account for environmental covariates, different time of data collection, different experimenters, etc. As you mention there might be some problems with using BLUPs, though the bias might be to underestimate the mean differences between lines. The estimates of the genetic covariance between traits and fitness are presented in Figure 4, from the G_qw_ matrix (all 6 transition rates, body size and fertility). These estimates, however, do not include assessing within-line variation in fertility. To address this problem we provide an additional analysis where we vary within-line fertility, to mimic the assay sampling. This is presented in Figure 4, figure supplement 1 (see also reply #20 above). We do not observe much bias.

c. L.246 the authors describe a null distribution, but how was this constructed? Morrissey et al. (2019; https://doi.org/10.1111/evo.13842) made an important suggestion for more conservative estimates of null G (also described in Hangartner et al. 2020; https://doi.org/10.1111/evo.13891).

Randomization of G, see reply #28. We referred to Morrissey et al. when performing the eigentensor analysis, which is now presented as an appendix in our GitHub repository. We did follow Morrissey et al. suggestion.

56. The description of the statistical analyses is sometimes difficult to follow. There are (by necessity) many parameters estimated and some more justification throughout the methods would help.a. Some clarity would help in the description of equations 1-3 as it is difficult to understand what the motivation is for each equation, or what they represent. Also, equations 4-5 could be easier to understand by including (in text) the definition of a plasticity vector from Noble et al. (2019): δ X = Xnn – Xnov, where X is the mean of each trait in the nonnovel and novel environments.

We are sorry for this. We have modified the phenotypic analysis and the different variables are now, hopefully, better described. We have also included a notation table (Table 1).

b. L.257 and Equation10-12: I think equation 10 represents the amount of genetic variance in the direction of plasticity, but as a proportion of total plasticity (i.e. the length of the plasticity vector), is this correct? If so, please clarify in the text.

It does not represent the genetic variance in the direction of plasticity as a proportion of total plasticity. The equation simply accounts for the fact that the plasticity vector might not be of length one. We have revised this whole section.

57. I am also confused by equations 10-11, why not just calculate the proportion of genetic variance in the direction of maximum evolvability (i.e. denominator in equation 10 is the 1st eigenvalue of G) – with plasticity vector normalised to unit length so that it represents the direction of plasticity. This removes the need for equation 11 and gives you the same information: the amount of genetics in the direction of plasticity as a proportion of the direction of maximum genetic variance. I apologise again if I've misunderstood, but I think a more detailed description and justification for equations 10-11 would help.

Thank you for this comment. We used the framework defined by Noble et al. PNAS 2019 for simplicity but we realized that more explanation was needed. It is not necessary to normalize by the largest eigenvalues as we only compare the empirical values to the null expectation. However, we think that the reader will better understand the metric if it scaled between 0 and 1.

c. Equation 12: I don't think this is the correct null expectation. Why would we expect plasticity to be in the direction of 'average' genetic variance? I would think a better null would be random vectors (of unit length) projected through G so that you would test whether plasticity is in a direction that describes greater genetic variance than expected by random sampling.

As you suggest we have eliminated equation 12, and now only present one null distribution (obtained from randomizing the G-matrix). Note that the projections are of the Gmatrix through the plasticity (or divergence) vectors, and not the other way around.

d. Equation 5: Is this calculated for each of the derived populations? This is important for understanding the results in Figure 5. Because they are independent populations (with G also estimated separately), to me it would make more sense to do pairwise comparisons for each derived population with the ancestral (which could be summarised in Figure 5).

Indeed, these are now what we call δ_q (Table 1). We estimate the difference between each evolved population relative to the ancestral population, using the MANOVA. The results are presented in supplementary data table 2 in Figure 6 (and illustrated with asterisks). We use δ_q, for each evolved replicate population, to estimate the genetic selection gradients (Figure 9).

e. Figure 6 and 7 (+ their interpretation): To test how to predict phenotypic evolution, I think it would be better to directly compare the distribution of β and the selection differentials. By comparing them separately it is not clear what hypothesis is being tested and the argument is verbal rather than quantitative – see Hajduk et al. (2020; https://doi.org/10.1098/rstb.2019.0359) for a nice example.

As suggested, we present a comparison between genetic and phenotypic selection gradients on each trait (Figure 9, panel A); obtained with Lande’s equation and assuming a stable G-matrix over multiple generations (CI are obtained by sampling the G-matrix posterior distribution, see replies above). Note that the estimates of phenotypic selection gradients have larger means and CI than the genetic selection gradients, leading to high replicate heterogeneity. For this reason we would like to continue to present the tests on the direction and magnitude of phenotypic evolution (Figure 9, panels B and C). All tests are quantitative even if not formally defined.

58. In the discussion, I found it very interesting that the authors found body size to be both genetically correlated to the movement traits, and that selection was predicted accurately (both sg and β). I am curious as to why this trait wasn't included in the analyses because, as the authors highlight, it is probably the trait selection is operating on (or at least provides an estimate of performance as outlined by the traditional Lande and Arnold selection analyses). It made me wonder how body size changed across treatments and whether it evolved in the high salt treatment. I would think that it would be important to include body size in the analysis.

We now include body size from the start of the analysis, with additional insights about the evolution of phenotypic plasticity and novelty. We have deleted the discussion about direct versus indirect selection.

59. The discussion could be a bit more concise. I also think alternative explanations need to be discussed as I'm not sure I agree with the interpretation on L.425. It is more parsimonious (or at least a viable alternative) that during adaptation genetic variance has been depleted as would be expected if the selection is strong. Another alternative would be that selfing in a stressful environment has reduced the amount of genetic variance. I think it could be worth including these alternative explanations.

We have tried to reduce and simplify the Discussion. We now include a paragraph on the loss of genetic variance by selection (and eliminated the discussion on overdominance). We also mention the expectations for the loss of genetic variance due to drift under selfing assuming infinitesimal trait inheritance.

[Editors' note: further revisions were suggested prior to acceptance, as described below.]

Reviewer #3 (Recommendations for the authors):The authors present an interesting test of whether experimental evolution can be predicted from the patterns of genetic variation in the ancestral population. The authors have done an impressive job to address the comments raised in the previous review, which means that the paper is much stronger and focused on the questions addressed. I appreciated their detailed responses to the questions raised in the previous review and I found the new version greatly improved.I only have one comment: The writing, while comprehensive, is quite dense and often abstract throughout the paper. This will make it difficult for a broader audience to follow. In addition, because of the (necessary) statistical rigor, it felt like the biology is missing, especially in the setup of the study and the results. For example, terms such as 'canonical traits' and references to alignments in the first paragraph make it difficult for a more broader audience to understand the background of the study, or the gap in knowledge that is being addressed.

We want to thank the reviewer for the constructive remarks. We tried in this last version of the manuscript to address all of your comments and suggestions. As surely you agree, explaining every technically in vague or unprecise terms is unfeasible. We further provide numerous references to reviews of the several methods and topics that we address.

I provide some specific examples below, but suggest that revising should focus on clarity throughout (and on the biological rather than statistical importance):– L29-33 it seems early in the introduction to introduce G and Lande's equation, and it does so with little biological foundation.

We have added two sentences with biological motivation before introducing Lande’s equation in a separate paragraph (line 25).

– L.13 a better way of saying 'follow their selection gradient' would be 'whether phenotypic evolution occurs in the direction of selection'.

Thank you, corrected.

– L.14 'the canonical trait of the multivariate phenotype' this is not broadly accessible. Perhaps 'trait combinations with large amounts of genetic variation' (or something similar) would be more intuitive.

We have re-phrased the sentence and now are explicitly defining what we mean by canonical traits, following your suggestion (line 39).

The premise of the study is much clearer but could be clarified further. The paragraph in the introduction L.115-116 needs to be simplified as it is difficult to follow and just lists the contrasts in a complicated way. Instead, it would be better to include the tests and hypotheses, rather than 'characterising' adaptation. The final two sentences are confusing (L.123-127), they should more clearly outline the focus of the study.

Thank you. We have greatly reduced this paragraph to just include the hypotheses being tested.

In particular, I'm not sure what the final sentence is saying: are you testing whether genetic selection gradients in the ancestral population = phenotypic selection gradients after adaptation? I would have thought you would compare genetic selection gradients with phenotypic divergence between the ancestral and adapted populations.

Please see also the reply below. Briefly, and following your suggestion in the first round of review, we used Hadjuk et al. framework to compare the genetic selection gradient with the phenotypic selection gradient, using in both circumstances Lande’s retrospective equation (which allows us to include G-matrix uncertainties). As mentioned in the reply below, this is different (but complementary) from the comparison of the selection differentials with the observed phenotypic divergence (which was presented in the first version of the manuscript, and continues to be so in Figures 9BC). Hajduk et al. are not explicit in testing for responses to selection when comparing phenotypic with genetic selection gradients (their equation 1.3). They do follow up with using genetic selection gradients to predict phenotypic evolution using Lande’s equation δ_Z = G times Β_g, but this is different from testing for indirect versus direct selection.

I liked the addition of the explanation of the traits in the introduction, but is it possible to add some more biology? L.111-112 '… which ones [traits] are genetically or environmentally independent' is a bit vague. Do you instead have an idea of how the movement traits help them to navigate low vs high salt environments? Or some other more biological reasoning. It is suggested 'movement can increase during experimental evolution due to more foraging and dwelling' – but this is confusing, is it that there is selection for greater movement across generations? Or just that there is greater movement in a new environment? Foraging and dwelling seem like opposites unless dwelling is defined as a specific behaviour.

We are not able to directly connect the change in our locomotion traits with changes in specific behaviors such as foraging, dwelling, sleep, mate search etc.. and we do not want the readers to believe that we can. We have specifically defined a phenotypic space that is general enough to encompass these behaviors without the necessity to describe them in anthropomorphized terms. After all, we only take locomotion behavior traits as a surrogate for any multivariate trait evolution. This said, for example, foraging would be something like preferring forward movement, and dwelling a preference for rapid changes in the direction of movement. We have added a sentence in the introduction to clarify the biological significance of locomotion behavior in our two environments and refer the reader to Fravell et al. 2020 where a discussion about the complexities of behavior locomotion can be found.

I found the major tests of indirect vs direct selection and also how phenotypic evolution is predicted very difficult to follow in the results. I think the analyses are correct, but the results need to be described more simply so that it is easier to understand. I'm not sure the section on direct selection is required, and if removed, could help simplify the study to focusing on predicting phenotypic evolution.

We would like to keep the section on indirect versus direct selection, as predicting multivariate trait evolution crucially depends on being aware of it. Please see next reply.

Or there needs to be a better justification for including direct selection because the Stinchombe (2014) and Hajduk et al. (2020) framework does not seem to have been used (but I apologise if I've misinterpreted something). Specifically, is there a reference for equation 7? Why does this Β represent phenotypic selection when it seems to capture phenotypic divergence. I am also a little confused as to why selection differentials are not just compared to observed divergence. And if you want to include direct vs indirect, predicted evolutionary change with indirect selection could be calculated using δ Z = G*Β where Β is calculated as per equation 6 – this uses the Stinchcombe (2014) framework. Sorry again if I've misinterpreted something, but there seems to be a disconnect between the results as written, the figures and the interpretation seems contradictory in the abstract (L.15-16).Overall I found the study very interesting, I hope my comments help.

Maybe we have not properly explained our analyses and results. Strictly, we have not used Stinchcombe et al. or Hajduk et al. approaches because they estimate the phenotypic selection gradients (Β) using the Lande-Arnold regression. Instead we use Lande’s retrospective equation to estimate both Β and Stinchcombe’s genetic selection gradients, Β_g. Lande’s retrospective equation (our equations 6 and 7) are equation 9 in Lande 1979, which is now referred to in the methods, line 706 – besides being presented in the introduction.

We interpreted your suggestion in the previous round of reviewing for us to apply Lande’s retrospective equation for Β because this is what is usually done in natural populations (as only divergence data are available) and your suggestion was on how to properly predict phenotypic evolution. As mentioned in the reply above, the explicit test for direct selection in Hajduk proposal is that Β be equal to Β_g, that is that the phenotypic covariance between trait and fitness (Lande-Arnold’s regression) over total phenotypic variance is equal to the genetic covariance between traits’ and fitness

(Robertson’s selection differentials) over the genetic variance for the trait. We agree that our approach appears to confound the question of direct versus indirect selection with predicting phenotypic evolution. However, the test for indirect versus direct selection (the comparison of Β and Β_g, Figure 9A), is presented separately from the test of predicting phenotypic evolution (comparison of observed divergence with selection differentials, Figures 9B and 9C). Overall, we agree that both tests appear to be confounded, here and in the literature.

We also note that our approach considers the uncertainty in the estimation of the ancestral G-matrix for both Β and Β_g (though admittedly, like R. Lande, we assume an unchanging G with evolution). In the paragraph on indirect versus direct selection, we refer to this issue (see also supplementary analysis to Figure 9). Our approach does not seem to have been reported before.